# A Unified Approach to Two-Dimensional Brinkman-Bénard Convection of Newtonian Liquids in Cylindrical and Rectangular Enclosures

**DOI:** 10.3390/e26010002

**Published:** 2023-12-19

**Authors:** Pradeep G. Siddheshwar, Kanakapura M. Lakshmi, David Laroze

**Affiliations:** 1Centre for Mathematical Needs, Department of Mathematics, CHRIST (Deemed to Be University), Hosur Road, Bengaluru 560029, India; pg.siddheshwar@christuniversity.in; 2Department of Mathematics, School of Physical Sciences, Central University of Karnataka, Kalaburagi 585367, India; 3Instituto de Alta Investigación, Universidad de Tarapacá, Casilla 7D, Arica 1000000, Chile; dlarozen@academicos.uta.cl

**Keywords:** Brinkman–Bénard convection, cylindrical and rectangular geometries, onset of convection

## Abstract

A unified model for the analysis of two-dimensional Brinkman–Bénard/Rayleigh–Bénard/ Darcy–Bénard convection in cylindrical and rectangular enclosures (
CE/RE
) saturated by a Newtonian liquid is presented by adopting the local thermal non-equilibrium (
LTNE
) model for the heat transfer between fluid and solid phases. The actual thermophysical properties of water and porous media are used. The range of permissible values for all the parameters is calculated and used in the analysis. The result of the local thermal equilibrium (
LTE
) model is obtained as a particular case of the 
LTNE
 model through the use of asymptotic analyses. The critical value of the Rayleigh number at which the entropy generates in the system is reported in the study. The analytical expression for the number of Bénard cells formed in the system at the onset of convection as a function of the aspect ratio, 
$S_{o}$
, and parameters appearing in the problem is obtained. For a given value of 
$S_{o}$
 it was found that in comparison with the case of 
LTE
, more number of cells manifest in the case of 
LTNE
. Likewise, smaller cells form in the 
DBC
 problem when compared with the corresponding problem of 
BBC
. In the case of 
RBC
, fewer cells form when compared to that in the case of 
BBC
 and 
DBC
. The above findings are true in both 
CE
 and 
RE
. In other words, the presence of a porous medium results in the production of less entropy in the system, or a more significant number of cells represents the case of less entropy production in the system. For small and finite 
$S_{o}$
, the appearance of the first cell differs in the 
CE
 and 
RE
 problems.

## 1. Introduction

Rayleigh–Bénard convection (
RBC
) in a fluid-saturated porous medium has been widely studied in the last few decades due to its wide range of applications in many fields, including thermal insulation systems, geothermal reservoirs, cooling of electronic and thermal equipment, etc. Generally, in nature, two types of porous media are seen, viz., high-porosity (loosely packed) and low-porosity (densely packed) media. Mathematically, loosely packed porous media are modelled using the Brinkman–Forchheimer-extended Darcy/Brinkman model. The densely packed porous medium is modelled using the Darcy model. These models consider the geometrical properties of the medium. Observing the thermal properties of the fluid and the porous medium, one adopts either the local thermal equilibrium (
LTE
) or the local thermal nonequilibrium (
LTNE
) model. When the difference exists in the thermal properties of fluid and solid phases, then the consideration of the 
LTNE
 model becomes necessary to analyse the system.

We first present a detailed literature survey concerning 
RBC
 in a loosely-packed porous medium (Brinkman–Bénard convection—
BBC
) followed by that of a densely-packed porous medium (Darcy–Bénard convection—
DBC
) occupying a rectangular geometry and a cylindrical geometry with the 
LTNE
 assumption. The articles studying 
BBC
 with 
LTNE
 [1,2,3,4,5,6,7,8] and 
DBC
 with 
LTNE
 [9,10,11,12,13,14] in rectangular enclosures (
RE
) are covered below. Phanikumar and Mahajan [1] examined the heat transfer characteristics of natural convection flow in a fluid-saturated metal-foam porous medium heated from below. They modelled the porous medium flow using the Brinkman–Forchheimer-extended Darcy model (
BFED
 model). They conclude that the 
LTNE
 model is a better model to study convection in the fluid-metal-foam porous medium. Postelnicu and Rees [2] performed a linear stability analysis of 
BBC
 of a Newtonian liquid using the 
LTNE
 model. They found excellent agreement between the asymptotic and the numerical results of heat transfer. Using the 
LTNE
 model, Straughan [3] determined the threshold value of convection at which instability occurs by making use of a global nonlinear stability analysis. He showed that the critical value predicted by the nonlinear theory exactly matches the one predicted by the linear theory. Khashan et al. [4] numerically simulated the natural convection problem in fluid-saturated porous cavities, including non-Darcian effects: Brinkman, convecting and Forchheimer terms. The effect of non-Darcian terms is well examined over a wide range of Rayleigh numbers. Postelnicu [5] made a linear stability analysis of 
DBC
 with 
LTNE
 assumption using single-term and *N*-term Galerkin methods. Using a single-term Galerkin method, an explicit expression is obtained for the Rayleigh number as a function of the wave number and other parameters. The eigenvalue obtained from this has been used as an initial guess value for the Galerkin method. Siddheshwar and Siddabasappa [6] analytically made linear and non-linear stability analyses of 
BBC
 using the 
LTNE
 model with rigid and free boundary conditions. The results of the Darcy model are obtained as a limiting case. Liu et al. [7] numerically investigated the flow structure and the heat transfer of 
BBC
 in a square lattice. They examined the effect of porosity on the flow properties during the transition of flow from 
BBC
 to 
DBC
. Siddheshwar et al. [8] analytically studied the 
BBC
 problem of four visco-elastic/one Newtonian fluid-saturated media occupying rectangular/cylindrical enclosures. They developed a common Lorenz model for both 
CE
 and 
RE
 to analyse the dynamics of the system. We next move on to the literature survey concerning 
DBC
–
LTNE
 problems.

Banu and Rees [9] numerically investigated the onset of 
DBC
 in a fluid-saturated porous medium using the two-temperature model between fluid and solid phases. The results of the 
LTE
 model are recovered by taking the thermal equilibrium limits (asymptotic analyses). Govender and Vadasz [10] performed a linear stability analysis of 
DBC
 with Coriolis and gravitational body forces and concluded that the effect of rotation is to stabilise convection even when the system exhibits thermomechanical anisotropy. Postelnicu [11] studied the effect of pressure gradient on the onset of 
DBC
 using the Galerkin method and the numerical solver 
dsolve
 of Maple. The results of the 
LTE
 model are recovered for large values of the scaled interface heat transfer parameter. Bidin and Rees [12] made a weakly non-linear analysis of the classical 
DBC
 problem to analyse the effect of the 
LTNE
 assumption on different planforms of convection. Secondary instabilities like Eckhaus and zig-zag instabilities were also considered. They showed that the roll planform is a stable planform of convection. Siddheshwar et al. [13] investigated both regular and chaotic convection in a densely-packed porous medium saturated by a Newtonian liquid by considering phase lag effects. A novel decomposition method is used to obtain an extended Vadasz-Lorenz model. They found from the study that the phase-lag effects alter the nature of chaos. Bansal and Suthar [14] analysed the onset of 
DBC
 using the 
LTNE
 model when the temperature of the two boundaries varies sinusoidally. The matrix differential operator theory is adopted to perform the stability analysis, and the critical Rayleigh number as a function of system parameters is obtained. The problem of 
RBC
 in 
RE
 is well investigated by many researchers and is now documented in books [15,16,17,18,19,20]. Kita [21] showed in his study the principle of maximum entropy as a general rule to study the stability of a 
RBC
 in non-equilibrium steady states. Jia et al. [22] studied the characteristics of 
RBC
 systems entropy production using the centre mesh systems and the finite volume method. They found from the study that the Rayleigh number and the aspect ratio play a crucial role in the formation of the Bénard cells pattern in the system. We next present the literature survey of 
CE
.

There is no reported literature on 
BBC
 with 
LTNE
, 
DBC
 with 
LTNE
 and 
BBC
 with 
LTE
 (except the work of Siddheshwar et al. [8], the study is restricted to unicellular convection) in a cylindrical geometry. Thus, we shall now focus our attention on the literature survey of 
DBC
 with 
LTE
 approximation in cylindrical enclosures (
CE
) and cylindrical annuli (
CA
) [23,24,25,26,27,28,29,30,31,32]. Zebib [23] analytically studied the onset of 
DBC
 in cylindrical enclosures. They found that the asymmetric mode is the preferred mode of convection except in the range of aspect ratio (radius/height) [1.09:1.28] at which the axisymmetric mode of convection occurs. Bau and Torrance [24] examined the onset of 
DBC
 in cylindrical annuli with two types of top wall boundary conditions, viz., permeable and impermeable. They showed that the system is more stable in the case of impermeable boundary condition compared to the permeable one. Haugen and Tyvand [25] performed a linear stability analysis of 
DBC
 with impermeable and conducting boundary conditions. They showed that the axisymmetric mode of convection is always a preferred one. In their analysis, they predicted the number of cells for some ranges of aspect ratios using the stream function plots. Zhang et al. [26] determined the critical Darcy–Rayleigh number for 
DBC
 in a cylindrical enclosure saturated by a visco-elastic fluid. They also determined the preferred mode of convection and found it to be a function of the aspect ratio and visco-elastic parameters. Bringedal et al. [27] examined the onset of 
DBC
 in a cylindrical porous annulus with impermeable, heat-conducting or insulating boundary conditions. They found that the effect of the inserted solid cylinder on the onset of convection is more dominant in the case of conducting boundary conditions compared to the insulating one. Kuznetsov and Nield [28] analytically investigated the onset of through-flow convection in a porous cylinder with vertical heterogeneity using a single-term Galerkin method. They found from the study that due to the symmetry of the horizontal boundary condition, the through flow is stabilising. Barletta and Storesletten [29] analysed the onset of convection in a densely packed vertical porous cylinder. The Robin boundary condition on temperature is applied to the horizontal boundaries, and the side boundary is assumed to be thermally insulated. The study shows that the Biot number plays an important role in determining the transition of the aspect ratio from one mode to the other. They also showed that the principle of exchange of stabilities is valid in the study. Kang et al. [30] analysed the onset of convection in a visco-elastic fluid-saturated rotating porous annulus. They examined the effect of the Taylor number and visco-elastic parameters on the onset and the mode of convection. Barletta and Storesletten [31] performed a linear stability analysis on 
DBC
 in a porous cylinder with permeable and thermally insulated vertical boundaries. Hence, the side wall is constrained by the temperature and pressure distributions. They first report results on 
DBC
 in a circular cylinder and then generalise the study to arbitrary cross-sectional cylinders. Siddheshwar and Lakshmi [32] analytically studied the onset and the heat transport by 
DBC
 in cylindrical enclosures and cylindrical annuli. They found from the study that the onset is advanced, and the heat transport is enhanced in the case of a cylindrical annulus compared to a cylindrical enclosure when the same volume is considered in the two geometries. The problem of 
RBC
 is one of the limiting cases of the 
BBC
 problem. Having completed a literature survey of 
RBC
 in 
CE
, we found that [33,34,35,36,37,38,39,40] have considered the problem.

From the literature survey on Newtonian fluid occupying a porous medium, we found that the following aspects have not yet been investigated:Analytical study of 
BBC
 in 
CE
 with 
LTE/LTNE
.Analytical study of 
RBC
 in 
CE
.Analytical study of 
DBC
 in 
CE
 with 
LTNE
.

In addition to the above-mentioned open problems, there are no reported results on aspects concerning the dependence of the number of cells and cell size on the aspect ratio, thermophysical properties and geometrical properties of the porous medium. Most importantly, explicit expression connecting cell size with the parameters representing these properties does not seem to be there. It is the endeavour of this paper to find such an expression and gain insight into the nature of cells that form. Also, our paper intends to report a unified analysis covering all the above-mentioned unconsidered problems in cylindrical and rectangular enclosures.

## 2. Mathematical Formulation

Analytical study of two-dimensional Rayleigh–Bénard convection in a fluid-saturated sparsely-packed porous medium confined in cylindrical/rectangular enclosures is considered for investigation. The considered porous medium is assumed to be isotropic and made up of spherically shaped porous solids. The lower and upper plates of the enclosures are assumed to be thin and maintained at temperatures 
$T_{0}+\Delta T$
 and 
$T_{0}$
, respectively, with 
ΔT>0
, and are separated by a distance *d*. The horizontal dimension of the enclosures is taken as *b*. The chosen coordinate system has *s* and *y* axes, respectively, along the radial/horizontal and vertical directions.

(1)
$$\frac{\partial u}{\partial s}+\frac{\delta }{\delta (1-s)+s}\frac{u}{s}+\frac{\partial v}{\partial y}=0,$$


(2)
$$\frac{\rho_{f}}{\phi }\frac{\partial \vec{Q}*}{\partial t}=-\nabla^{*}P+\mu_{e}\nabla^{{*}^{2}}\vec{Q}*-\frac{\mu_{f}}{K}\vec{Q}*+\left[\rho_{f}-\left(\rho_{f}\beta \right)\left(T_{f}-T_{0}\right)\right]\vec{g},$$


(3)
$$\left(\rho_{f}c_{f}\right)\left[\phi \frac{\partial T_{f}}{\partial t}+\left(\vec{Q}*\cdot \nabla^{*}\right)T_{f}\right]=\phi k_{f}\nabla^{{*}^{2}}T_{f}+h\left(T_{s}-T_{f}\right),$$


(4)
$$\left(\rho_{s}c_{s}\right)\left(1-\phi \right)\frac{\partial T_{s}}{\partial t}=k_{s}\left(1-\phi \right)\nabla^{{*}^{2}}T_{s}-h\left(T_{s}-T_{f}\right),$$

where 
$\vec{Q}*=u\left(s,\;y,\;t\right)\hat{e_{s}}+v\left(s,\;y,\;t\right)\hat{e_{y}}$
 is the two-dimensional velocity vector, 
δ
 is the curvature parameter (artificially introduced) which takes two values 0 and 1 corresponding to the problems of cylindrical and rectangular enclosures, respectively. The quantities 
$\phi =\frac{\mathrm{Volume}\;\mathrm{of}\;\mathrm{pores}}{\mathrm{Volume}\;\mathrm{of}\;(\mathrm{pores}\;+\;\mathrm{porous}\;\mathrm{matrix})}$
 is the porosity, 
$K=\frac{\phi^{3}d_{s}^{2}}{180{\left(1-\phi \right)}^{2}},\;d_{s},\;\rho ,\;c,\;\mu ,\;t,P,$

β,g,T,k
 and *h* are, respectively, permeability, diameter of the solid spheres (porous medium), density, specific heat at constant pressure, dynamic viscosity, time, pressure, thermal expansion coefficient, acceleration due to gravity, temperature, thermal conductivity and heat transfer coefficient. The operator 
$\nabla^{*}=\hat{e_{s}}\frac{\partial }{\partial s}+\hat{e_{y}}\frac{\partial }{\partial y}$
 is the two-dimensional gradient operator. The subscripts 
f,s,e,and0
, respectively, represent the quantities pertaining to fluid, solid, effective quantities (in terms of fluid and solid phases’ values) and reference value. The reference value is taken at 
$T=T_{0}$
.

The pressure term in Equation (Equation 2) will now be eliminated by operating curl twice, and then on simplifying, we obtain the equation for the *y*-component of velocity as

(5)
$$\frac{\rho_{f}}{\phi }\frac{\partial }{\partial t}\left(\nabla^{{*}^{2}}v\right)=\mu_{e}\nabla^{{*}^{4}}v-\frac{\mu_{f}}{K}\nabla^{{*}^{2}}v+\left(\rho_{f}\beta \right)g\nabla_{s}^{2}T_{f},$$

where 
$\nabla^{{*}^{2}}=\nabla_{s}^{2}+\frac{\partial^{2}}{\partial y^{2}}$
 is the Laplacian operator and 
$\nabla_{s}^{2}=\frac{\partial^{2}}{\partial s^{2}}+\frac{\delta }{s}\frac{\partial }{\partial s}$
 is the horizontal Laplacian operator.

The dimensional quantities present in Equations (Equation 1)–(Equation 4) are non-dimensionalised using the following quantities:
(6)
$$\begin{array}{cc} \; & \tau =\frac{\chi_{f}}{d^{2}}t,\;\left(S,\;Y\right)=\left(\frac{s}{b},\;\frac{y}{d}\right),\;\left(U,\;V\right)=\frac{d}{\phi \chi_{f}}\left(u,\;v\right),\;\left(\Theta ,\;\Phi \right)=\frac{\left(T_{f},\;T_{s}\right)}{\Delta T}, \end{array}$$

where 
$\chi =\frac{k}{\rho c}$
 is the thermal diffusivity.

On non-dimensionalising Equations (Equation 1)–(Equation 4) using Equation (Equation 6), we obtain:
(7)
$$\frac{1}{S_{o}}\frac{\partial U}{\partial S}+\frac{\delta }{S_{o}\left[\delta (1-S)+S\right]}\frac{U}{S}+\frac{\partial V}{\partial Y}=0,$$


(8)
$$\frac{1}{Pr}\frac{\partial }{\partial \tau }\left(\nabla^{2}V\right)=\Lambda \nabla^{4}V-\sigma^{2}\nabla^{2}V+Ra\nabla_{S}^{2}\Theta ,$$


(9)
$$\frac{\partial \Theta }{\partial \tau }=\nabla^{2}\Theta -\left(\vec{Q}\cdot \nabla \right)\Theta +H\left(\Phi -\Theta \right),$$


(10)
$$\alpha \;\frac{\partial \Phi }{\partial \tau }=\nabla^{2}\Phi -H\gamma \left(\Phi -\Theta \right).$$

The non-dimensional parameters present in Equations (Equation 7)–(Equation 10) are defined as follows: (Brinkman [41])

(11)
$$\left.\begin{array}{cc} Pr=\frac{\phi \mu_{f}}{\rho_{f}\chi_{f}}\;\left(\begin{array}{c} \mathrm{porosity}-\mathrm{modified}\\ \mathrm{Prandtl}\;\mathrm{number} \end{array}\right),\;\;\; & \alpha =\frac{\chi_{f}}{\chi_{s}}\;\begin{array}{c} (\mathrm{thermal}\;\mathrm{diffusivity}\\ \;\mathrm{ratio}) \end{array},\\ H=\frac{hd^{2}}{\phi k_{f}}\;\left(\begin{array}{c} \mathrm{scaled}\;\mathrm{inter}-\mathrm{phase}\\ \mathrm{heat}\;\mathrm{transfer}\;\mathrm{coefficient} \end{array}\right), & \Lambda =\frac{\mu_{e}}{\mu_{f}}\;\left(\mathrm{Brinkman}\;\mathrm{number}\right),\\ \gamma =\frac{\phi k_{f}}{\left(1-\phi \right)k_{s}}\;\left(\begin{array}{c} \mathrm{porosity}-\mathrm{weighted}\\ \mathrm{thermal}\;\mathrm{conductivity}\;\mathrm{ratio} \end{array}\right), & \sigma^{2}=\frac{d^{2}}{K}\;\left(\mathrm{porous}\;\mathrm{parameter}\right),\\ Ra=\frac{\rho_{f}\beta g\Delta Td^{3}}{\phi \mu_{f}\chi_{f}}\;\left(\begin{array}{c} \mathrm{porosity}-\mathrm{modified}\\ \mathrm{thermal}\;\mathrm{Rayleigh}\;\mathrm{number} \end{array}\right), & S_{o}=\frac{b}{d}\;(\mathrm{aspect}\;\mathrm{ratio}),\\ \mu_{e}=\frac{\mu_{f}}{\phi^{2.5}}\;. & \end{array}\right.$$


The operators appearing in Equations (Equation 7)–(Equation 10) are:
(12)
$$\left.\begin{array}{cccc} \; & \nabla_{S}^{2}=\frac{1}{S_{o}^{2}}\frac{\partial^{2}}{\partial S^{2}}+\frac{1}{S_{o}^{2}}\frac{\delta }{S}\frac{\partial }{\partial S}, & \nabla^{2} & =\frac{1}{S_{o}^{2}}\frac{\partial^{2}}{\partial S^{2}}+\frac{1}{S_{o}^{2}}\frac{\delta }{S}\frac{\partial }{\partial S}+\frac{\partial^{2}}{\partial Y^{2}}. \end{array}\right.$$

At the quiescent basic state, the system is in an entropy equilibrium condition due to the assumption that the destabilising temperature variations are compensated by the stabilising effect (caused by the viscosity) of the fluid-saturated porous medium. Hence, we assume that the heat is transported by conduction alone, and there is local thermal equilibrium (
LTE
) between the phases. Thus, we take

(13)
$$\vec{Q}=0,\;\;\Theta =\Theta_{b}\left(Y\right),\;\;\Phi =\Phi_{b}\left(Y\right)\;\;\mathrm{and}\;\;\Theta_{b}=\Phi_{b}.$$

With the Equation (Equation 13), the governing Equations (Equation 7)–(Equation 10) reduce to:
(14)
$$\frac{d^{2}\Theta_{b}}{dY^{2}}=0,\;\;\;\frac{d^{2}\Phi_{b}}{dY^{2}}=0.$$

The basic state Equation (Equation 14) is solved subject to the following conditions:
(15)
$$\begin{array}{c} \Theta =\Theta_{0}+1,\;\;\Phi =\Phi_{0}+1\;\;\mathrm{at}\;\;Y=0\;\;\mathrm{and}\;\;0<S<1,\\ \Theta =\Theta_{0},\;\;\Phi =\Phi_{0}\;\;\mathrm{at}\;\;Y=1\;\;\mathrm{and}\;\;0<S<1 \end{array}$$

and the resultant solution is:
(16)
$$\Theta_{b}=1-Y,\;\;\;\Phi_{b}=1-Y.$$


Now, to check the stability of the system, we superimpose a small perturbation on the quiescent basic state, which generates entropy in the system, as:
(17)
$$\vec{Q}=\vec{Q^{\prime }},\;\;\Theta =\Theta_{b}+\Theta^{\prime },\;\;\Phi =\Phi_{b}+\Phi^{\prime },$$

where prime denotes a perturbed quantity. Using Equation (Equation 17) in the governing Equations (Equation 7)–(Equation 10) and then using Equation (Equation 16) in the resulting equations, we obtain:
(18)
$$\frac{1}{S_{o}}\frac{\partial U^{\prime }}{\partial S}+\frac{\delta }{S_{o}\left[\delta (1-S)+S\right]}\frac{U^{\prime }}{S}+\frac{\partial V^{\prime }}{\partial Y}=0,$$


(19)
$$\frac{1}{Pr}\frac{\partial }{\partial \tau }\left(\nabla^{2}V^{\prime }\right)=-\sigma^{2}\nabla^{2}V^{\prime }+\Lambda \nabla^{4}V^{\prime }+Ra\;\nabla_{S}^{2}\Theta^{\prime },$$


(20)
$$\frac{\partial \Theta^{\prime }}{\partial \tau }=\nabla^{2}\Theta^{\prime }-\left(\vec{q^{\prime }}\cdot \nabla \right)\Theta^{\prime }+V^{\prime }+H\left(\Phi^{\prime }-\Theta^{\prime }\right),$$


(21)
$$\alpha \frac{\partial \Phi^{\prime }}{\partial \tau }=\nabla^{2}\Phi^{\prime }-H\gamma \left(\Phi^{\prime }-\Theta^{\prime }\right).$$

Further on, in Equations (Equation 18)–(Equation 21), we neglect primes for simplicity and solve them subject to the following boundary conditions:

Impermeable, stress-free and isothermal horizontal boundaries

(22)
$$V=0,\;\mu_{e}\left[\frac{\partial U}{\partial Y}+\frac{\partial V}{\partial S}\right]=0,\;\Theta =0,\;\Phi =0\;\;\mathrm{on}\;Y=0,\;1\;\mathrm{and}\;0<S<1.$$


Impermeable, stress-free and adiabatic vertical boundaries

(23)
$$V=0,\;\mu_{e}\left[\frac{\partial V}{\partial S}+\frac{\partial U}{\partial Y}\right]=0,\;\frac{\partial \Theta }{\partial S}=0,\;\frac{\partial \Phi }{\partial S}=0\;\mathrm{on}\;S=0,\;1\;\mathrm{and}\;0<Y<1.$$

On simplifying Equations (Equation 22) and (Equation 23) using the continuity Equation (Equation 18), we obtain

(24)
$$\left.\begin{array}{cc} V=\frac{\partial^{2}V}{\partial Y^{2}}=0,\;\;\Theta =0,\;\;\Phi =0\;\;\mathrm{on}\;\;Y=0,\;1\;\;\; & \mathrm{and}\;0<S<1,\\ U=\nabla_{S}^{2}U=0,\;\;\frac{\partial \Theta }{\partial S}=0,\;\;\frac{\partial \Phi }{\partial S}=0\;\;\mathrm{on}\;\;S=0,\;1\;\;\; & \mathrm{and}\;0<Y<1, \end{array}\right.$$

where the boundary 
S=0
 in the case of 
CE
 is a pseudo-boundary.

Having obtained the nondimensional form of governing equations and boundary conditions, we now present the linear stability analysis to determine the critical Rayleigh number at which convection occurs.

## 3. Linear Stability Analysis under the Assumption of the Principle of Exchange of Stabilities

On linearising Equations (Equation 19)–(Equation 21) and considering the steady state, we obtain: 
(25)
$$\begin{array}{c} -\sigma^{2}\nabla^{2}V+\Lambda \nabla^{4}V+Ra\nabla_{S}^{2}\Theta =0, \end{array}$$

(26)
$$\begin{array}{c} \nabla^{2}\Theta +V+H\left(\Phi -\Theta \right)=0, \end{array}$$

(27)
$$\begin{array}{c} \nabla^{2}\Phi -H\gamma \left(\Phi -\Theta \right)=0. \end{array}$$

The 
Y−
dependent part of the variable separable eigenfunctions corresponding to the periodically appearing roll planform (velocity) and for isothermal conditions (fluid and solid temperatures) are of the form 
sin[πY]
. Thus, we take

(28)
$$V\left(S,\;Y\right)=V^{*}\left(S\right)\sin\left[\pi Y\right],\;\Theta \left(S,\;Y\right)=\Theta^{*}\left(S\right)\sin\left[\pi Y\right],\;\Phi \left(S,\;Y\right)=\Phi^{*}\left(S\right)\sin\left[\pi Y\right].$$

The solution (Equation 28) satisfies *Y*-boundary conditions given in Equation (Equation 24). In view of this, for the purpose of calculation, we may now write 
$\nabla_{S}^{2}-\pi^{2}$
 for 
$\nabla^{2}$
. With this, Equations (Equation 25)–(27) now become: 
(29)
$$\begin{array}{c} \Lambda \nabla_{S}^{4}V^{*}-\left(\sigma^{2}+2\pi^{2}\Lambda \right)\nabla_{S}^{2}V^{*}+\pi^{2}\left(\sigma^{2}+\pi^{2}\Lambda \right)V^{*}+Ra\nabla_{S}^{2}\Theta^{*}=0, \end{array}$$

(30)
$$\begin{array}{c} \left(\nabla_{S}^{2}-\pi^{2}\right)\Theta^{*}+V^{*}+H\left(\Phi^{*}-\Theta^{*}\right)=0, \end{array}$$

(31)
$$\begin{array}{c} \left(\nabla_{S}^{2}-\pi^{2}\right)\Phi^{*}-H\gamma \left(\Phi^{*}-\Theta^{*}\right)=0. \end{array}$$

Decoupling of 
$V^{*},\;\Theta^{*}$
 and 
$\Phi^{*}$
 from Equations (Equation 29)–(31) and the resulting equations are unified as follows: 
(32)
$$\begin{array}{c} \nabla_{S}^{8}\varpi^{*}-\left[q_{1}+\frac{q_{2}}{\Lambda }+2\pi^{2}\right]\nabla_{S}^{6}\varpi^{*}+\left[\left(\pi^{2}+q_{1}\right)\left(2\pi^{2}+\frac{q_{2}}{\Lambda }\right)+\frac{\sigma^{2}\pi^{2}-Ra}{\Lambda }\right]\nabla_{S}^{4}\varpi^{*}\\ +\left[\frac{Ra}{\Lambda }q_{3}-\pi^{4}\left(q_{1}+\frac{q_{2}}{\Lambda }\right)-2\pi^{2}q_{1}\frac{q_{2}}{\Lambda }\right]\nabla_{S}^{2}\varpi^{*}+\frac{\pi^{4}}{\Lambda }q_{1}q_{2}\varpi^{*}=0, \end{array}$$

where

(33)
$$\varpi^{*}={\left[V^{*}\;\Theta^{*}\;\Phi^{*}\right]}^{T},\;\;q_{1}=\pi^{2}+H\left(1+\gamma \right),\;\;q_{2}=\sigma^{2}+\pi^{2}\Lambda ,\;\;q_{3}=\pi^{2}+H\gamma .$$

To solve Equation (Equation 32), we need 8 boundary conditions with respect to *S* on each of 
$V^{*},\;\Theta^{*}$
 and 
$\Phi^{*}$
. Equation (Equation 24) gives us only two boundary conditions on 
Θ
 and 
Φ
 and four boundary conditions on *U*. But we need boundary conditions only on *V*, and hence we convert *U* boundary conditions present in Equation (Equation 24) into *V* boundary conditions using Equation (Equation 18). This procedure gives us:
(34)
$$\frac{\partial V^{*}}{\partial S}=\frac{\partial }{\partial S}\left(\nabla_{S}^{2}V^{*}\right)=0\;\;\mathrm{on}\;\;S=0,\;1\;\;\;\mathrm{and}\;0<Y<1.$$

At this point, we are short of four and six boundary conditions on 
$V^{*}$
 and 
$\left(\;\Theta^{*},\;\Phi^{*}\right)$
, respectively, with respect to *S*. The additional boundary conditions required are obtained using Equations (Equation 29)–(31) and the available boundary conditions. The required boundary conditions on adding new ones on 
$V^{*},\;\Theta^{*}$
 and 
$\Phi^{*}$
 are: 
(35)
$$\frac{\partial \varpi^{*}}{\partial S}=\frac{\partial }{\partial S}\left(\nabla_{S}^{2}\varpi^{*}\right)=\frac{\partial }{\partial S}\left(\nabla_{S}^{4}\varpi^{*}\right)=\frac{\partial }{\partial S}\left(\nabla_{S}^{6}\varpi^{*}\right)=0\;\;\mathrm{on}\;\;S=0,\;1\;\;\mathrm{and}\;\;0<Y<1.$$

Let us now factorise Equation (Equation 32) as follows:
(36)
$$\left(\nabla_{S}^{2}+m^{2}\right)\left(\nabla_{S}^{2}+n^{2}\right)\left(\nabla_{S}^{2}+p^{2}\right)\left(\nabla_{S}^{2}+q^{2}\right)\varpi^{*}=0,$$

where 
$m^{2},\;n^{2},\;p^{2}$
 and 
$q^{2}$
 are to be determined. On multiplying together the four factors in Equation (Equation 36), we obtain

(37)
$$\begin{array}{c} \nabla_{S}^{8}\varpi^{*}+\left(m^{2}+n^{2}+p^{2}+q^{2}\right)\nabla_{S}^{6}\varpi^{*}+\left(m^{2}n^{2}+m^{2}p^{2}+m^{2}q^{2}+n^{2}p^{2}+n^{2}q^{2}+p^{2}q^{2}\right)\nabla_{S}^{4}\varpi^{*}\\ +\left(m^{2}n^{2}p^{2}+m^{2}n^{2}q^{2}+n^{2}p^{2}q^{2}+m^{2}p^{2}q^{2}\right)\nabla_{S}^{2}\varpi^{*}+m^{2}n^{2}p^{2}q^{2}\varpi^{*}=0. \end{array}$$

Now on comparing Equations (Equation 32) and (Equation 37), we obtain the following relations connecting 
$m^{2},\;n^{2},\;p^{2}$
 and 
$q^{2}$
: 
(38)
$$\begin{array}{cc} & m^{2}+n^{2}+p^{2}+q^{2}=-\left[q_{1}+\frac{q_{2}}{\Lambda }+2\pi^{2}\right], \end{array}$$

(39)
$$\begin{array}{cc} & m^{2}\left(n^{2}+p^{2}+q^{2}\right)+n^{2}p^{2}+n^{2}q^{2}+p^{2}q^{2}=\left[\left(\pi^{2}+q_{1}\right)\left(2\pi^{2}+\frac{q_{2}}{\Lambda }\right)+\frac{\sigma^{2}\pi^{2}-Ra}{\Lambda }\right], \end{array}$$

(40)
$$\begin{array}{cc} & m^{2}n^{2}p^{2}+m^{2}n^{2}q^{2}+n^{2}p^{2}q^{2}+m^{2}p^{2}q^{2}=\left[\frac{Ra}{\Lambda }q_{3}-\pi^{4}\left(q_{1}+\frac{q_{2}}{\Lambda }\right)-2\pi^{2}q_{1}\frac{q_{2}}{\Lambda }\right], \end{array}$$

(41)
$$\begin{array}{cc} & m^{2}n^{2}p^{2}q^{2}=\frac{\pi^{4}}{\Lambda }\left(q_{1}q_{2}-2Ra\right). \end{array}$$

In the above equations, with the intention of retaining only 
$m^{2}$
 we rewrite Equations (Equation 38), (39) and (41) as

(42)
$$\begin{array}{c} n^{2}+p^{2}+q^{2}=-\left[q_{1}+\frac{q_{2}}{\Lambda }+2\pi^{2}\right]-m^{2}, \end{array}$$


(43)
$$\begin{array}{c} n^{2}p^{2}+n^{2}q^{2}+p^{2}q^{2}=\left[\left(\pi^{2}+q_{1}\right)\left(2\pi^{2}+\frac{q_{2}}{\Lambda }\right)+\frac{\sigma^{2}\pi^{2}-Ra}{\Lambda }\right]-m^{2}\left(n^{2}+p^{2}+q^{2}\right), \end{array}$$


(44)
$$\begin{array}{c} n^{2}p^{2}q^{2}=\frac{\pi^{4}}{\Lambda }\left(q_{1}q_{2}-2Ra\right)/m^{2}. \end{array}$$

We next simplify Equation (40) as:
(45)
$$\begin{array}{c} m^{2}\left(n^{2}p^{2}+n^{2}q^{2}+p^{2}q^{2}\right)+n^{2}p^{2}q^{2}=\left[\frac{Ra}{\Lambda }q_{3}-\pi^{4}\left(q_{1}+\frac{q_{2}}{\Lambda }\right)-2\pi^{2}q_{1}\frac{q_{2}}{\Lambda }\right]. \end{array}$$

On using Equations (Equation 42)–(44) in the Equation (Equation 45), we obtain

(46)
$$\begin{array}{c} {\left(m^{2}\right)}^{4}+\left[q_{1}+\frac{q_{2}}{\Lambda }+2\pi^{2}\right]{\left(m^{2}\right)}^{3}+\left[\left(\pi^{2}+q_{1}\right)\left(2\pi^{2}+\frac{q_{2}}{\Lambda }\right)+\frac{\sigma^{2}\pi^{2}-Ra}{\Lambda }\right]{\left(m^{2}\right)}^{2}\\ -\left[\frac{Ra}{\Lambda }q_{3}-\pi^{4}\left(q_{1}+\frac{q_{2}}{\Lambda }\right)-2\pi^{2}q_{1}\frac{q_{2}}{\Lambda }\right]m^{2}+\frac{\pi^{4}}{\Lambda }\left(q_{1}q_{2}-2Ra\right)=0. \end{array}$$

The Equation (Equation 46) may be used to solve for 
m
, but it involves the eigenvalue 
Ra
. In view of this we rearrange Equation (Equation 46) to obtain the expression for 
Ra
 in the form:
(47)
$$Ra=\frac{{\left(m^{2}+\pi^{2}\right)}^{2}}{m^{2}}\left[\Lambda \left(m^{2}+\pi^{2}\right)+\sigma^{2}\right]\left[1+\frac{H}{m^{2}+\pi^{2}+H\gamma }\right].$$


The parameter 
Ra
 is an eigenvalue which characterises the production of entropy in the system once it crosses its threshold value. We call it a critical Rayleigh number, 
$Ra_{c}$
.

At this point, we shift our attention to obtaining the solution of 
V,Θ
 and 
Φ
 of Equations (Equation 25)–(27). In what follows, we shall obtain the solution for the Equation (Equation 36) for 
$\varpi^{*}$
. We hence choose the Helmholtz Equation:
(48)
$$\left(\nabla_{S}^{2}+m^{2}\right)\varpi^{*}=0.$$

The solution of the Helmholtz Equation in (Equation 48) is:
(49)
$$\left.\begin{array}{c} \varpi^{*}=AS^{\frac{1-\delta }{2}}J_{\frac{\delta -1}{2}}\left[mS_{o}S\right], \end{array}\right.$$

where 
$A={\left[A_{0}\;B_{0}\;C_{0}\right]}^{T}$
 is the matrix of infinitesimal amplitudes of convection, 
$J_{\frac{\delta -1}{2}}\left[mS_{o}S\right]$
 is the Bessel function of the first kind and of order 
$\frac{\delta -1}{2}$
. The solution in Equation (Equation 49) is used in Equation (Equation 28) to obtain the complete solution of 
$\varpi ={\left[V\;\Theta \;\Phi \right]}^{T}$
 in the form:
(50)
$$\;\begin{array}{c} \varpi =AS^{\frac{1-\delta }{2}}J_{\frac{\delta -1}{2}}\left[mS_{o}S\right]\sin\left[\pi Y\right]. \end{array}$$


Equation (Equation 50) satisfy the boundary condition in Equation (Equation 35), provided:
(51)
$$J_{\frac{1+\delta }{2}}\left[mS_{o}\right]=0.$$

From the above proceeding it becomes evident that the critical 
m
, viz., 
$m_{c}$
, is not just the result of the minimisation of 
Ra
 with respect to 
m
. It also involves the constraint condition (Equation 51). Thus, we have on hand a constraint minimisation problem involving Equations (Equation 47) and (Equation 51). To simplify the constraint minimisation problem further, we make the substitution 
$a=mS_{o}$
 in Equations (Equation 47) and (Equation 51). With this substitution, Equations (Equation 47) and (Equation 51) now take the form:
(52)
$$Ra=\frac{{\left(a^{2}/S_{o}^{2}+\pi^{2}\right)}^{2}}{a^{2}/S_{o}^{2}}\left[\Lambda \left(a^{2}/S_{o}^{2}+\pi^{2}\right)+\sigma^{2}\right]\left[1+\frac{H}{a^{2}/S_{o}^{2}+\pi^{2}+H\gamma }\right],$$


(53)
$$J_{\frac{1+\delta }{2}}\left[a\right]=0.$$


There are infinitely more solutions for 
a
 that satisfy the condition (Equation 53). Among these values of 
a
, a particular value which minimises the Rayleigh number is the required critical value of 
a
, namely 
$a_{c}$
.

After having evolved the right procedure for obtaining the critical values of 
a
 and thereby 
$m_{c}$
 and 
$Ra_{c}$
 in the case of the unified problem, we next discuss the results obtained in the study.

## 4. Results and Discussion

The present article reports various problems of axisymmetric 
CE
 and two-dimensional 
RE
 in a unified way. Various problems unified into one problem shall be explained in brief now. More details on the limiting cases shall be given later on in the section. Brinkman–Bénard convection (
BBC
) of Newtonian liquids in axisymmetric 
CE
 and two-dimensional 
RE
 is studied analytically(by artificially introducing the curvature parameter) using the two-temperature model adopted for the 
LTNE
 situation. When the thermal properties of the two phases (fluid and solid) are quite different, then the 
LTNE
 assumption is required to be considered. In such a case, the two-temperature model will have to be used to study the thermo-fluid dynamics. This situation is all the more true when an enormous temperature is observed in the system. However, when the temperature involved is not that high, then the temperatures of the two phases do not differ much when the values of thermophysical quantities are nearly the same. This situation is termed as 
LTE
. One can mathematically obtain the results of the 
LTE
 model from those of the 
LTNE
 model by making asymptotic analyses [2,6,9]. In the present analysis, we have considered a loosely packed porous medium. By neglecting factors that characterise the porous medium, we can study the results of the clear fluid case. With the intention of obtaining the results of a densely-packed porous medium, we may consider small permeability and, hence very large values of the porous parameter.

From the above, it is clear that several problems can be unified into a single problem (see Figure 1). In what follows, we present the following problems (limiting cases) in essential detail:

Two-dimensional 
BBC
 problems in 
CE
 and 
RE
 with 
LTNE
.Two-dimensional 
DBC
 problems in 
CE
 and 
RE
 with 
LTNE
.Two-dimensional 
RBC
 problems in 
CE
 and 
RE
.Two-dimensional 
BBC
 problems in 
CE
 and 
RE
 with 
LTE
.Two-dimensional 
DBC
 problems in 
CE
 and 
RE
 with 
LTE
.

As a result of the unified handling, we shall be dealing with ten limiting cases as one and the same is shown in Figure 1.

In the present paper, the material of the porous medium is such that its thermophysical properties are much different from those of the fluid occupying it. Hence, of relevance to this paper is the 
LTNE
 situation only. However, for academic completeness, we consider the 
LTE
 situation too and arrive at some results. The very general formulation adopted here involves a loosely packed porous medium that allows us to handle both the no porous medium and the densely packed porous medium cases.

To make the results have a practical value, we have made use of the actual thermophysical properties of the fluid and of the porous matrix. The details are provided in the next subsection.

### 4.1. Thermophysical Properties and the Parameters’ Values concerning a Fluid-Saturated Porous Medium

The fluid and porous media considered in the analysis are water, glass fibre (
GF
), sand, glass balls (
GB
) and aluminium foam (
AF
) porous medium, and their thermophysical properties are present in the articles [42,43]. Among the four porous media considered, 
GF
 has high porosity; hence, we use that for a specific investigation concerning 
BBC
. The other three porous media have low porosity; hence, we use them in the study of 
DBC
. Using the thermophysical properties mentioned in Table 1, we shall now calculate various parameters, 
Pr,α,Λ,γ,H
 and 
$\sigma^{2}$
 appearing in Section 2.

The parameter 
$Pr=\frac{\phi \mu_{f}}{\rho_{f}\chi_{f}}$
 is the porosity-modified Prandtl number. Since 
Pr
 is a function of 
ϕ
, it is found to be different for 
GF(ϕ=0.88)
 and other porous media (
ϕ=0.5
). The permissible value of 
Pr
 for different water-saturated porous media is calculated, and the same is recorded in the first column of Table 1. The diffusivity ratio, 
$\alpha =\frac{\chi_{f}}{\chi_{s}}$
, is calculated and its values are recorded in the second column of Table 1. The values of 
$\gamma =\frac{\phi k_{f}}{\left(1-\phi \right)k_{s}}$
 are computed for four different fluid-saturated porous media and are recorded in the third column of Table 1. The parameter, 
$\Lambda =\frac{\mu_{e}}{\mu_{f}}$
, is the ratio of effective viscosity to the viscosity of the fluid. The effective viscosity is calculated using the phenomenological law, 
$\mu_{e}=\frac{\mu_{f}}{\phi^{2.5}}$
 and 
Λ
 values are documented in the fourth column of Table 1.

The four parameters discussed above are dependent only on the thermal properties of the fluid and porous media and do not depend on the height of the enclosures. The parameters 
$H=\frac{hd^{2}}{\phi k_{f}}$
 and 
$\sigma^{2}=\frac{d^{2}}{K}$
 are clearly dependent on the height of the enclosure, where 
$K=\frac{\phi^{3}d_{s}^{2}}{180{\left(1-\phi \right)}^{2}}$
 is the permeability of porous media. In the paper, we choose two different heights (
d=0.02
 m, 
0.03
 m) of enclosures and the obtained parameters’ values that depend on *d* are tabulated in Table 2. In the experimental work of Liang et al. [33], they mention that asymmetric convection is observed if the height of the enclosure is greater than 
0.03
 m. Since our study concerns axisymmetric convection, we have chosen 
d≤0.03
 m.

To find the permissible value for the parameter *H*, it is required to find the heat transfer coefficient, *h*. Kuwahara et al. [44] reported a numerical study to find a correlation for *h* as a function of 
Pr
, Reynold’s number, 
$Re=\frac{\rho_{e}d\;V}{\mu_{e}}$
, and 
ϕ
, where 
$\rho_{e}=\phi \rho_{f}+\left(1-\phi \right)\rho_{s}$
 is the effective density. The fluid flow velocity, *V*, in the cases of high-porosity and low-porosity media, is chosen as 
0.01
 m/s and 
0.001
 m/s, respectively. The expression of *h* from Kuwahara et al. [44] is given by

(54)
$$h=\frac{k_{f}}{d_{s}}\left[\left(1+\frac{4(1-\phi )}{\phi }\right)+\frac{1}{2}\sqrt{\left(1-\phi \right)}Re^{0.6}Pr^{1/3}\right].$$

Also, *h* depends on the diameter of the solid spheres, 
$d_{s}$
. Here, we have chosen the value of 
$d_{s}$
 in the range of 
[0.005:0.02]
 m and 
[0.001:0.005]
 m, respectively, for the cases of loosely and densely-packed porous media. The notation 
[a:b]
 implies that the quantity takes the least value of *a* and the maximum value of *b*. Now, using Equation (Equation 54), we found the permissible range for *H* for the considered porous media and the same is tabulated in Table 2. The parameters 
K
 and 
$\sigma^{2}$
 are found for the above-mentioned range of values of 
$d_{s}$
, and the same are also tabulated in Table 2.

Having discussed the thermophysical properties and having estimated the non-dimensional parameters, we now consider the limiting cases: 
RBC
 and 
DBC
 and obtain the expression of the Rayleigh number in the two cases.

### 4.2. Expressions of the Rayleigh Number in the Cases of 
RBC
 and 
DBC


We first consider the case of 
RBC
. In the case of no porous medium, the value of 
ϕ
 is 1 and, hence, 
$\mu_{e}=\mu_{f}$
. For this value of 
ϕ
 and 
$\mu_{e}$
, 
K
 is infinite and hence the parameters 
$\sigma^{2}$
 and 
Λ
 take the values 0 and 1, respectively. Now the Rayleigh number expression (Equation 52) in the case of a clear fluid takes the form:
(55)
$$Ra=\frac{{\left(a^{2}/S_{o}^{2}+\pi^{2}\right)}^{3}}{a^{2}/S_{o}^{2}}.$$


We next consider the 
DBC
 problem. From the recorded values of 
$\sigma^{2}$
 in Table 2, one can observe that the value of 
$\sigma^{2}$
 is very large in the case of densely-packed porous media (sand, 
GB
 and 
AF
) compared to the finite value of 
$\sigma^{2}$
 in the case of a loosely-packed porous medium(
GF
). When 
$\sigma^{2}\gg 1$
, then clearly 
$\Lambda \left(a^{2}/S_{o}^{2}+\pi^{2}\right)\ll \sigma^{2}$
 and hence the second term in 
$\sigma^{2}+\Lambda \left(a^{2}/S_{o}^{2}+\pi^{2}\right)$
 may be neglected.

The Equation (Equation 52) now takes the form:
(56)
$$Ra_{D}=\frac{{\left(a^{2}/S_{o}^{2}+\pi^{2}\right)}^{2}}{a^{2}/S_{o}^{2}}\left[1+\frac{H}{a^{2}/S_{o}^{2}+\pi^{2}+H\gamma }\right],$$

where 
$Ra_{D}=\frac{Ra}{\sigma^{2}}$
 is the Darcy–Rayleigh number.

After obtaining 
Ra
 expressions in the cases of 
BBC
, 
RBC
 and 
DBC
, in the next subsection, we shall consider asymptotic analyses to obtain the results of the 
LTE
 from the 
LTNE
.

### 4.3. Asymptotic Analyses: 
LTNE
 to 
LTE


The results of 
LTE
 can be extracted from those of 
LTNE
 using asymptotic analyses. This is possible in four different ways, as illustrated by [2,6,9]. The routes to 
LTE
 from 
LTNE
 are:
(57)
$$\left.\begin{array}{ccc} \; & \left(i\right) & H\to 0,\\ \; & \left(\mathrm{ii}\right) & H\to \infty ,\\ \; & \left(\mathrm{iii}\right) & \gamma \to \infty ,\\ \; & \left(\mathrm{iv}\right) & H\gamma \to \infty . \end{array}\right.$$

The result of extensive asymptotic analyses carried out by Postelnicu and Rees [2] concerns the cases of 
BBC
 and 
DBC
. The expression of the Rayleigh number of the 
LTE
 case that emerges from an asymptotic analysis in the cases of 
BBC
 and 
DBC
 is documented in Table 3.

The expression of the Rayleigh number for 
BBC
 and 
DBC
 clearly indicate that 
Ra
 is a function of 
a
 and 
$S_{o}$
. In the following section, we determine, through the constraint condition, the value of 
a
 that minimises 
Ra
 for a given value of 
$S_{o}$
.

### 4.4. General Solution of the Constraint Condition: 
$J_{\frac{1+\delta }{2}}\left[a\right]$
 = 0

We find the roots of the constraint condition in the cases of five problems, each of 
CE
 and 
RE
, that we have unified.

For 
δ=0
, the constraint takes the form:
(58)
sin[a]=0.

The roots of Equation (Equation 58) are:
(59)
a=nπ,n=1,2,⋯.

For 
δ=1
, the constraint simplifies to the form:
(60)
$$J_{1}\left[a\right]=0.$$

The Equation (Equation 60) yields discrete values as the solution. Let us name them as 
$a_{0},\;a_{1},\;a_{2},$

$a_{3},\cdots$
, where 
$a_{0}<a_{1}<a_{2}<a_{3}<\cdots$
. The first seven values of 
a
(for 
δ=0
 and 
δ=1
) are reported in Table 4. From the above, it is clear that 
a
 has an explicit expression for 
RE
 whereas 
a
 has discrete values only in 
CE
, and an explicit expression does not occur as in the case of 
RE
. In attempting to construct an explicit expression for 
a
 in 
CE
, we considered 100 discrete values of 
a
, viz., 
$a_{0},\;a_{1},\;a_{2},\cdots ,a_{99}$
. Making a close observation of these values, we found that 
$|a_{n+1}-a_{n}|\approx 3.1416\left(\approx \pi \right)$
 for 
n≥2
. Making use of this idea to find an explicit expression for 
a
 we tried out different linear interpolation functions: 
$a_{0}+n\pi ,\;a_{1}+\left(n-1\right)\pi ,\;a_{2}+\left(n-2\right)\pi ,\cdots a_{5}+\left(n-5\right)\pi$
, and found 
$a_{2}+\left(n-2\right)\pi$
 to be the line of best fit. Hence, we chose the following:
(61)
$$a=a_{2}+\left(n-2\right)\pi ,\;\;n=1,\;2,\;3\cdots .$$


In order to find a unified expression for 
a
 in five problems each of 
CE
 and 
RE
, we now write Equations (Equation 59) and (Equation 61) in unified form as:
(62)
$$a=(a_{2}-2\pi )\delta +n\pi ,\;\;n=1,\;2,\cdots .$$

We need to mention here that as yet, it is not clear as to what *n* and 
$m=\frac{a}{S_{o}}$
 represent physically. The answer to this shall, however, be provided in a succeeding section using plots of the stream function, but first, we shall discuss how to find 
$Ra_{c}$
.

### 4.5. Expression for the Critical Rayleigh Number

The substitution of Equation (Equation 62) in the Rayleigh number expression (Equation 52) gives us: 
(63)
$$\begin{array}{c} Ra\left(n,\;S_{o}\right)=\frac{{\left[{\left(\delta \left(a_{2}-2\pi \right)+n\pi \right)}^{2}/S_{o}^{2}+\pi^{2}\right]}^{2}}{{\left(\delta \left(a_{2}-2\pi \right)+n\pi \right)}^{2}/S_{o}^{2}}\times \\ \left[\Lambda \left[{\left(\delta \left(a_{2}-2\pi \right)+n\pi \right)}^{2}/S_{o}^{2}+\pi^{2}\right]+\sigma^{2}\right]\left[1+\frac{H}{{\left(\delta \left(a_{2}-2\pi \right)+n\pi \right)}^{2}/S_{o}^{2}+\pi^{2}+H\gamma }\right]. \end{array}$$

In finding the critical Rayleigh number, 
$Ra_{c}$
, after which entropy generates in the system, for each value of 
$S_{o}$
 we found that 
n=1
 is not necessarily the value which yields 
$Ra_{c}$
. The value 
n=1
 yields the minimum Rayleigh number until the aspect ratio crosses the threshold value, 
$S_{o}=2.17$
 in 
CE
. At 
$S_{o}=2.17$
, 
${\left(Ra_{c}\right)}_{n=1}={\left(Ra_{c}\right)}_{n=2}$
, after which 
n=2
 yields 
$Ra_{c}$
. Similarly after 
$S_{o}=3.50$
, 
n=3
 in 
CE
 results minimum 
$Ra_{c}$
.

This observation is graphically shown in Figure 2. For different ranges of 
$S_{o}$
, ascending integer values of *n* produces 
$Ra_{c}$
. The intersection points of piecewise continuous curves shown in Figure 2 actually represent the points of increase in the value of *n* (we have as yet not confirmed that *n* is the number of cells manifesting). To analyse this information from the physics point of view, we shall plot the stream function in the next subsection.

### 4.6. Streamlines and Physical Interpretation of n and 
a


Introducing velocity components *U* and *V* in terms of the stream function 
Ψ(S,Y)
, we obtain

(64)
$$U=-\frac{S_{o}\left[\delta \left(1-S\right)+S\right]}{S}\frac{\partial \Psi }{\partial Y},\;\;\;\;V=\frac{\left[\delta (1-S)+S\right]}{S}\frac{\partial \Psi }{\partial S}.$$

The velocity components *U* and *V* satisfy the continuity Equation (Equation 18). We have an explicit solution for 
V(S,Y)
, and hence, we use this in obtaining an expression for the stream function to obtain

(65)
$$\Psi =A_{0}\sin\left[\pi Y\right]\int \frac{S^{\frac{3-\delta }{2}}}{\delta (1-S)+S}J_{\frac{\delta -1}{2}}\left[a_{n}S\right]dS.$$

The integral constant in Equation (Equation 65) is zero since the boundary is a streamline. In Figure 3, we have plotted the stream function for a 
CE
 by considering one representative value of 
$S_{o}$
 between two intersecting points in Figure 2a.

Now, on observing Figure 2a and Figure 3 together, we find that intersection points in Figure 2 represent the points at which the number of cells increases. Quite obviously, we may now infer that *n* must represent the number of cells for a given 
$S_{o}$
. Similar arguments can be made for 
RE
 by observing Figure 2b and Figure 4. From this observation in Figure 2, Figure 3 and Figure 4, we can find sufficient information to obtain an explicit expression for the number of cells in the cases of 
CE
 and 
RE
. The details are given in the next section.

### 4.7. Explicit Expression for the Number of Cells in 
CE
 and 
RE


The intersection points of different modes (different *n*) in Figure 2 can be obtained by equating the critical Rayleigh number expressions of succeeding participating curves at their intersecting points. We thus take:
(66)
$$Ra\left(n,\;S_{o}\right)=Ra\left(n+1,\;S_{o}\right),$$

where 
$S_{o}$
 can now be only an intersection point. Now, using Equation (Equation 63) in Equation (Equation 66), we obtain an explicit expression for *n* as a function of 
$S_{o}$
 and other parameters appearing in the problem, in the form:
(67)
$$n=\frac{-\pi +\sqrt{\pi^{2}+4P}}{2\pi }+\left[2-\frac{a_{2}}{\pi }\right]\delta ,$$

where *P* is the real and positive root (only one, in fact) of the fifth-degree polynomial given by

(68)
$$\frac{q_{4}}{2}\left(\pi^{2}+q_{3}\;S_{o}^{2}\right)S_{o}^{8}+q_{4}\;S_{o}^{8}x+\left(q_{5}\;S_{o}^{2}+q_{6}\;S_{o}^{6}\right)x^{2}+q_{7}\;S_{o}^{2}x^{3}+q_{8}x^{4}+2\Lambda x^{5}.$$

The various quantities in the above polynomial are given by:
(69)
$$\begin{array}{c} q_{4}=-2\pi^{4}q_{1}\;q_{2},\\ q_{5}=\pi^{4}q_{3}\Lambda +q3\pi^{2}\left[q_{2}+\Lambda \left(q_{1}+2\pi^{2}\right)\right]S_{o}^{2},\\ q_{6}=\pi^{2}\left[q_{3}\sigma^{2}-q_{1}\left(\pi^{2}\Lambda +q_{2}\right)\right]+\left(q_{1}+\pi^{2}\right)\left[q_{3}\left(2\pi^{2}\Lambda +q_{2}\right)-\pi^{2}q_{2}\right],\\ q_{7}=4\pi^{2}\Lambda q_{3}+2q_{3}\left[\Lambda \left(q_{1}+2\pi^{2}\right)+q_{2}\right]S_{o}^{2},\\ q_{8}=\pi^{2}\Lambda +\left[\Lambda \left(q_{1}+2\pi^{2}\right)+q_{2}+3\Lambda q_{3}\right)S_{o}^{2}. \end{array}$$

The ceiling value of *n* represents the number of cells in the case of 
BBC
 with 
LTNE
 assumption. The ceil function returns the maximum integer value. The expression of *n* in the limiting cases of the present study is documented in Table 5.

The various quantities in Table 5 are given by

(70)
$$\begin{array}{c} q_{9}={\left[-1-9\left(1+3\;S_{o}^{2}-3\;S_{o}^{4}\right)\;S_{o}^{2}+i\;6\sqrt{3(1+9\;S_{o}^{2}+27\;S_{o}^{4})}\;S_{o}^{3}\right]}^{\frac{1}{3}},\;i=\sqrt{-1},\\ q_{10}=1+3\;S_{o}^{2},\;q_{11}=\frac{1}{3}\left\{-2q_{16}+3q_{3}^{2}S_{o}^{4}+{\left(\frac{2}{q_{17}}\right)}^{\frac{1}{3}}\left[q_{16}^{2}-12\pi^{6}q_{1}S_{o}^{6}\right]\right\},\\ q_{12}=\frac{1}{3}{\left(\frac{q_{17}}{2}\right)}^{\frac{1}{3}},\;q_{13}=\frac{1}{3}\left\{2\left(-2q_{16}+3q_{3}^{2}S_{o}^{4}\right)-{\left(\frac{2}{q_{17}}\right)}^{\frac{1}{3}}\left[q_{16}^{2}-12\pi^{6}q_{1}S_{o}^{6}\right]\right\},\\ q_{14}=2\left[\left(q_{3}^{2}-2\pi^{2}q_{3}+\pi^{4}\right)\left(q_{1}-q_{3}\right)+\pi^{4}q_{1}\right]S_{o}^{6}+2\pi^{2}q_{3}^{2}S_{o}^{4},\\ q_{14}=2\left[\left(q_{3}^{2}-2\pi^{2}q_{3}+\pi^{4}\right)\left(q_{1}-q_{3}\right)+\pi^{4}q_{1}\right]S_{o}^{6}+2\pi^{2}q_{3}^{2}S_{o}^{4},\\ q_{15}=2\sqrt{q_{13}-q_{12}+\frac{q_{14}}{\sqrt{q_{11}+q_{12}}}},\;q_{16}=\left[q_{1}q_{3}+2\pi^{2}\left(q_{3}-q_{1}\right)-\pi^{4}\right]S_{o}^{4}+\pi^{2}q_{3}S_{o}^{2},\\ q_{17}=2q_{16}^{3}+36\pi^{4}q_{16}q_{1}\left[2q_{18}+q_{3}S_{o}^{2}\right]S_{o}^{6}+\sqrt{q_{19}}+108\pi^{4}q_{1}\left[\pi^{4}q_{1}S_{o}^{2}-q_{18}q_{3}^{2}\right]S_{o}^{10},\\ q_{18}=\pi^{2}+q_{3}S_{o}^{2}, \end{array}$$


(71)
$$\begin{array}{c} q_{19}=4{\left[q_{16}^{3}+18\pi^{4}q_{1}q_{16}\left[2q_{18}+q_{3}S_{o}^{2}\right]S_{o}^{6}+54\pi^{4}q_{1}\left(\pi^{4}q_{1}S_{o}^{2}-q_{3}^{2}q_{18}\right)S_{o}^{10}\right]}^{2}\\ \;-4{\left[q_{16}^{2}-12\pi^{6}q_{1}S_{o}^{6}\right]}^{3},\\ q_{20}=\pi^{2}\Lambda +\left(3\pi^{2}\Lambda +\sigma^{2}\right)S_{o}^{2},\\ q_{21}=2\Lambda^{3}\pi^{6}\left(-1-9S_{o}^{2}-27S_{o}^{4}+27S_{o}^{6}\right)+6\pi^{4}\Lambda^{2}\sigma^{2}\left(-1-6S_{o}^{2}+9S_{o}^{4}\right)S_{o}^{2},\\ \;-6\pi^{2}\Lambda \sigma^{4}q_{10}S_{o}^{4}-2\sigma^{6}S_{o}^{6},\\ q_{22}={\left[q_{21}+\sqrt{-4q_{20}^{6}+q_{21}^{2}}\right]}^{\frac{1}{3}},\;q_{23}=\delta \left[2-\frac{a_{2}}{\pi }\right]. \end{array}$$

We note here that the quantity 
$\left[q_{10}\left(1-\frac{q_{10}}{q_{9}}\right)-q_{9}\right]$
 appearing in Table 5 for the case of 
RBC
 evaluates to a real value with a very small imaginary part even though 
$q_{9}$
 is a complex quantity.

Having obtained the expression for the number of cells in all possible cases, in the next section, we shall find the physical meaning of the parameter 
m
.

### 4.8. Physical Interpretation of the Parameter 
m


To analyse the parameter 
m
, we refer to Table 6, for water and four different water-saturated porous media considered wherein number of cells that shall appear for a given value of 
$S_{o}$
 is documented in the cases of 
CE
 and 
RE
. The corresponding cell sizes are calculated, and the same is recorded in Table 7.

From the tabulated values, we found that for small 
$S_{o}$
, the size of the cells is not the same in 
CE
 and 
RE
. However, for large values of 
$S_{o}$
, these are the same for 
CE
 and 
RE
. This means that the boundary effects are negligible at large 
$S_{o}$
, and we observe a uniformity in the cells. Hence, in this situation, the concept of wave number applies. This result is true for 
BBC
 with 
LTNE
 assumption and also in all its limiting cases.

Now recalling the definition of 
$m=\frac{a}{S_{o}}$
, we may write

(72)
$$m_{c}=\frac{\delta (a_{2}-2\pi )+n\pi }{S_{o}}.$$

Here, *n* may be replaced by the expression of the number of cells given by Equation (Equation 67) for 
BBC
–
LTNE
, and similarly, the expression of *n* of all limiting cases of 
BBC
–
LTNE
 documented in Table 5 can be used. Intuition tells us that the parameter 
m
 represents the wave number. To have confidence in this concept, we computed the value of 
$m_{c}=\frac{\delta (a_{2}-2\pi )+n\pi }{S_{o}}$
 for 
$S_{o}$
 in the range [10:100] for 
CE
 and 
RE
. The computations reveal that in both 
CE
 and 
RE
 cases, the value of 
$m_{c}$
 converges to the classical wave number of 
$\frac{\pi }{\sqrt{2}}$
. Hence, from now on, we shall refer to 
m
 as the wave number. Next, let us consider the stream function, 
Ψ
, in the range 
$\left[0,\;\frac{\delta a_{2}+2\pi }{a_{c}}\right]$
 in Figure 5 and Figure 6 for 
CE
 and 
RE
.

For 
CE
, we have chosen the range 
$\left[0,\;\frac{a_{2}+2\pi }{a_{c}}\right]$
 due to the existence of the stagnant region in the first cell compared to all other cells. From Figure 5, it is clear that the range 
$\left[0,\;\frac{a_{2}}{a_{c}}\right]$
 represents the initial wavelength, including the stagnant region, and the range 
$\left[\frac{a_{2}}{a_{c}},\;\frac{2\pi }{a_{c}}\right]$
 represents the actual wavelength. In Figure 6, we shall see one wavelength or two counter-rotating cells of 
RE
.

Now, some general observations made from the results of the study for any large 
$S_{o}$
 are:
$n^{RBC}<n^{BBC/DBC},\;\;m_{c}^{RBC}<m_{c}^{BBC/DBC}$
,
$n^{BBC}<n^{DBC},\;\;m_{c}^{BBC}<m_{c}^{DBC}$
,
$n^{LTNE}>n^{LTE},\;\;m_{c}^{LTNE}>m_{c}^{LTE}$
,
$n^{d=0.02m}>n^{d=0.03m},\;\;m_{c}^{d=0.02m}>m_{c}^{d=0.03m}$
.

Additional observations made from the tabulated values of Table 6 and Table 7 are as follows. In the case of 
LTE
, the cell size depends on the permeability of the porous medium and not on its thermophysical properties. Hence, we observe that the cell size is the same for 
DBC−LTE
 involving the three porous materials considered. This result is also true for 
BBC−LTE
. However, in the case of the corresponding problem under the 
LTNE
 assumption, we note that the cell size depends on not only the permeability but also the thermophysical properties of the porous medium. In that way, the results of 
BBC
 and 
DBC
 are different in so far as cell size is concerned when we consider the 
LTNE
 assumption. Another point to be observed is that the porous material with the largest thermal conductivity supports a smaller cell size.

Having obtained the expressions for the number of cells in different problems, we have recorded expressions of 
$Ra_{c}$
 and *n* of all the problems (including limiting cases) in Table 8.

The quantities mentioned in Table 8 are present in Equations (Equation 33), (Equation 69), (Equation 70) and (Equation 71). Having so far discussed the results of the present paper, in the next section we draw some major conclusions of the study.

## 5. Conclusions

Two-dimensional analyses of Brinkman–Bénard convection of Newtonian liquid in cylindrical and rectangular geometries are made using the two-temperature model. In the present model, ten individual problems are unified into a single problem:
BBC
 with 
LTNE
 in 
CE
 and 
RE
;
BBC
 with 
LTE
 in 
CE
 and 
RE
;
RBC
 in 
CE
 and 
RE
;
BBC
 with 
LTE
 in 
CE
 and 
RE
;
DBC
 with 
LTE
 in 
CE
 and 
RE
.

One of the important results is that given the value of the aspect ratio and parameters’ values, we can exactly predict how many cells manifest (in all 10 possible cases). This is without seeking recourse to streamline plots. A number of cells seen in 
RBC
 are found to be less when compared with the corresponding problems in a porous medium, i.e., 
$n^{RBC}<n^{BBC}<n^{DBC}$
. This result is true in both 
LTNE
 and 
LTE
 and also for 
CE
 and 
RE
. On comparing corresponding problems involving 
LTNE
 and 
LTE
 problems, we may write: 
$n^{LTNE}>n^{LTE}$
. For large aspect ratio, 
$S_{o}$
, the value of the wave number converges to 
$\frac{\pi }{\sqrt{2}}$
 in both 
CE
 and 
RE
, i.e., the vertical boundaries effect becomes negligible at large 
$S_{o}$
. This result coincides with the well-known result of a classical 
RBC
 problem, which validates the present model. However, the number of cells that manifest may not be the same for the corresponding problems in 
CE
 and 
RE
, even for the case of 
$S_{o}\gg 1$
.

Work is in progress to consider the experimental boundary conditions similar to the idealistic boundary conditions considered in this paper. Liang et al. [33] experimentally observed the circular roll patterns in a 
CE
 bounded by rigid boundaries. This justifies the consideration of circular roll patterns in the present analysis in the case of 
CE
. At this stage, the exact comparison of the present model with the experimental works is not possible as the present model considers the idealistic boundary conditions. The present study provides a qualitative picture of 
RBC
 in 10 different enclosures. The work on realistic boundary conditions is in progress. 

## Figures and Tables

**Figure 1 entropy-26-00002-f001:**
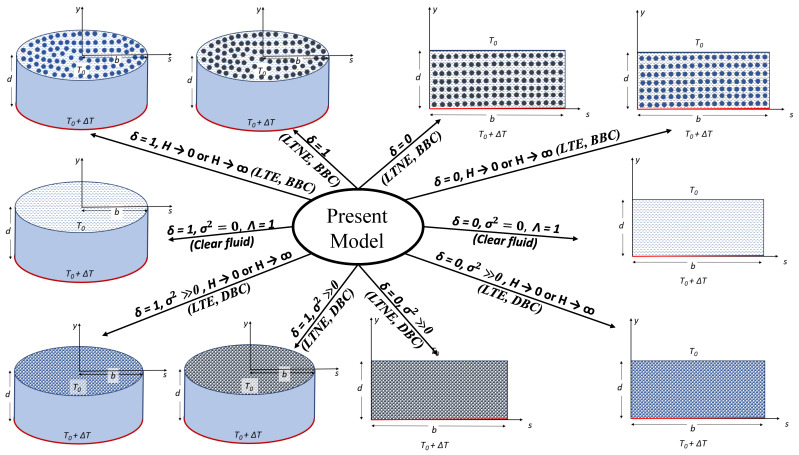
Schematic representation of the present model.

**Figure 2 entropy-26-00002-f002:**
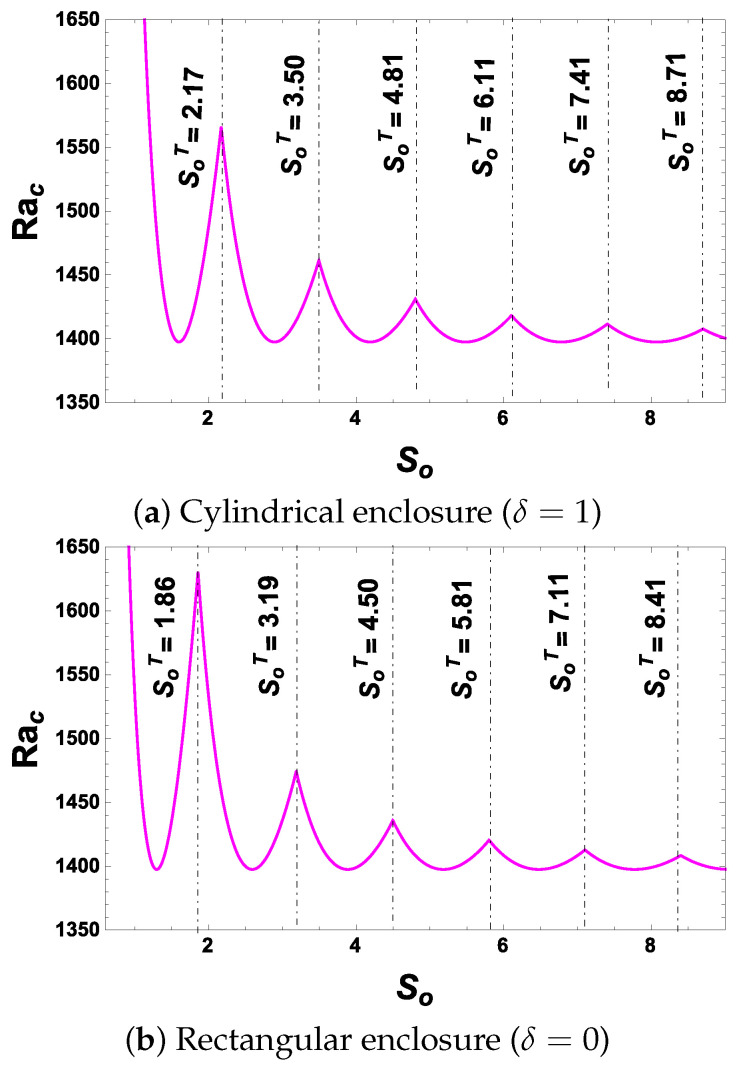
Plot of the critical Rayleigh number, 
$Ra_{c}$
, versus the aspect ratio, 
$S_{o}$
, for 
H=5,γ=18.7306,Λ=1.37655
 and 
$\sigma^{2}=10$
 values. 
$S_{o}^{T}$
 refer to values of 
$S_{o}$
 at intersection points.

**Figure 3 entropy-26-00002-f003:**
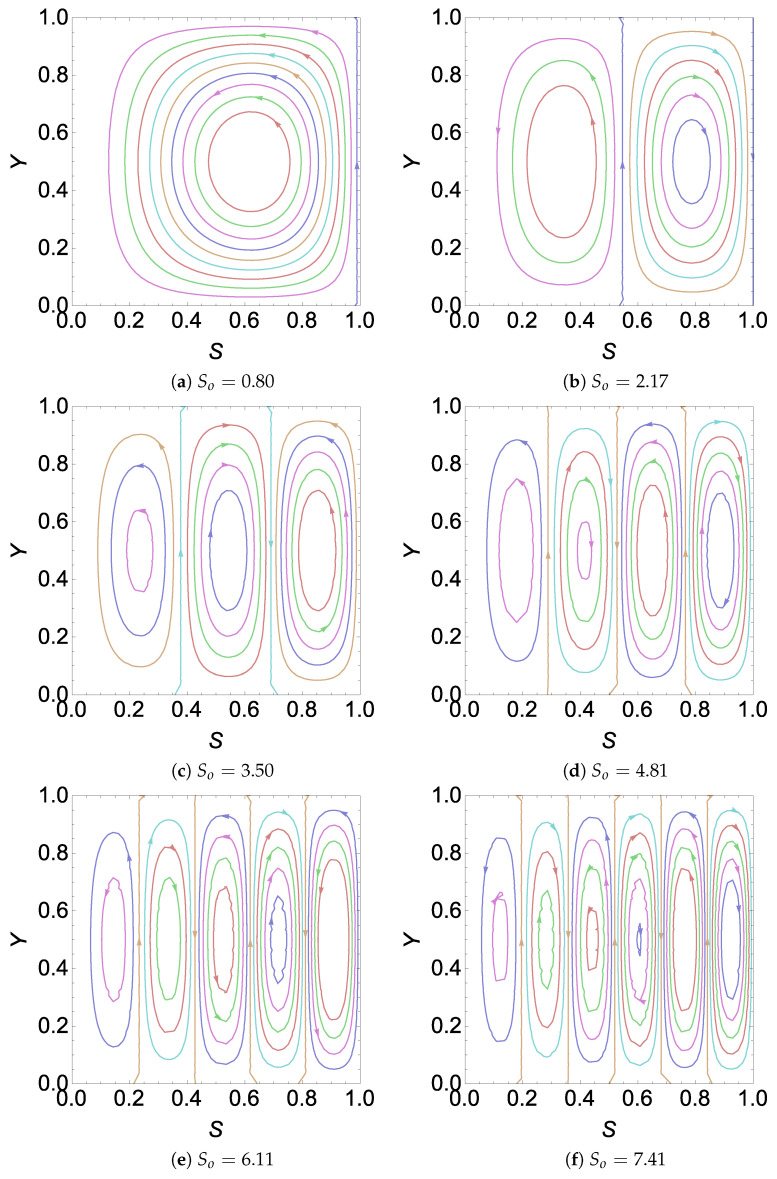
Plot of streamlines, 
Ψ
, for different values of 
$S_{o}$
 with 
H=5,γ=18.7306,Λ=1.37655
 and 
$\sigma^{2}=10$
 in a 
CE
.

**Figure 4 entropy-26-00002-f004:**
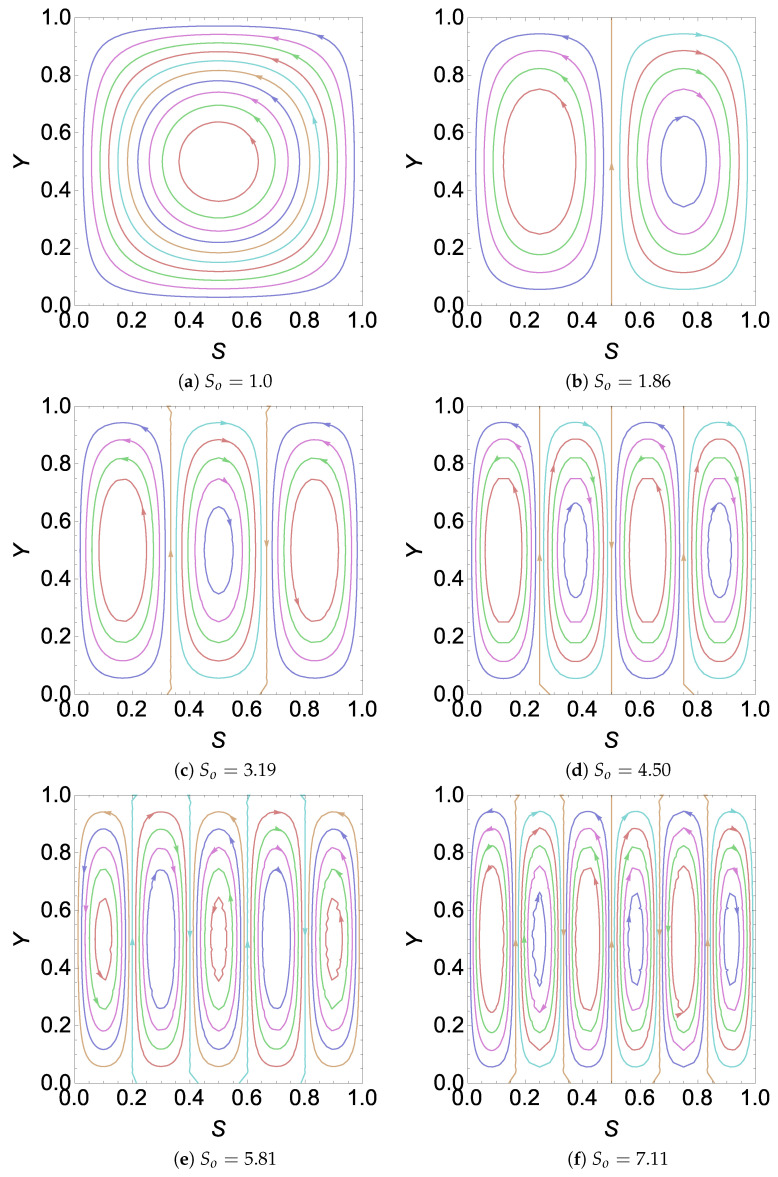
Plot of 
Ψ
 for different values of 
$S_{o}$
 with 
H=5,γ=18.7306,Λ=1.37655
 and 
$\sigma^{2}=10$
 in a 
RE
.

**Figure 5 entropy-26-00002-f005:**
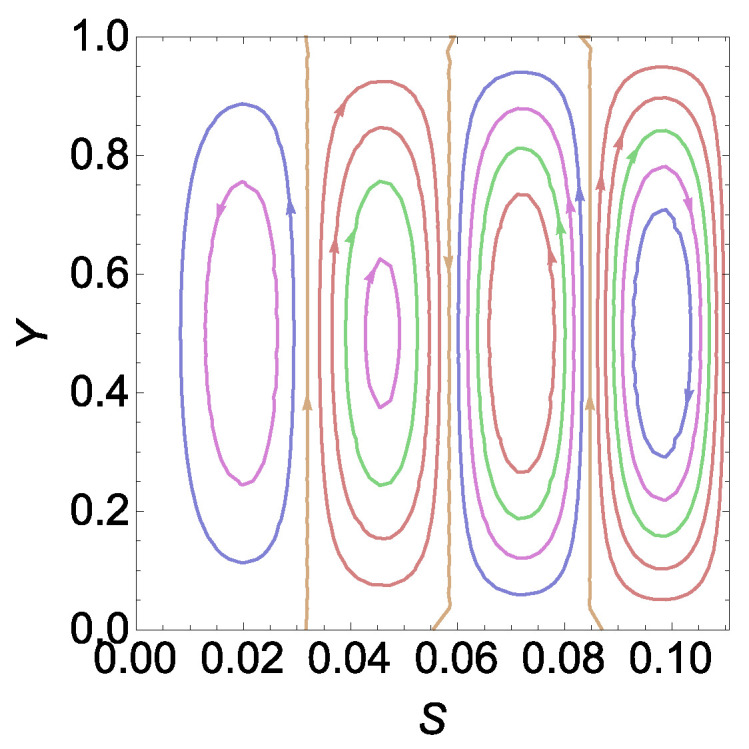
Plot of 
Ψ
 in the region 
$\left[0,\;\frac{a_{2}+2\pi }{a_{c}}\right]$
 in a 
CE
.

**Figure 6 entropy-26-00002-f006:**
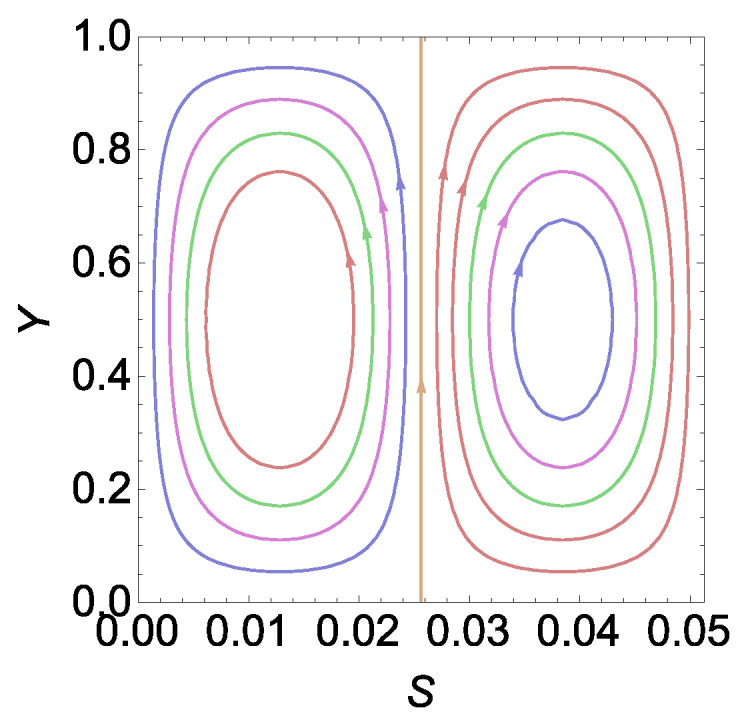
Plot of 
Ψ
 in the region 
$\left[0,\;\frac{2\pi }{a_{c}}\right]$
 in a 
RE
.

**Table 1 entropy-26-00002-t001:** The values of parameters 
(Pr,α,γ,Λ)
 for water-
GF
, water-Sand, water-
GB
 and water-
AF
.

	Pr	α	γ	Λ
water- GF	5.3393	0.9904	18.7306	1.37655
water-Sand	3.0336	0.6603	2.2703	5.6568
water- GB	3.0337	0.3177	0.5838	5.6568
water- AF	3.0337	0.0017	0.0030	5.6568

**Table 2 entropy-26-00002-t002:** The values of parameters *H* and 
$\sigma^{2}$
 for water-
GF
, water-Sand, water-
GB
 and water-
AF
.

	d(m)	*H*	$K(m^{-2})$	$\sigma^{2}$
water- GF	0.02	[0.5111:2.0446]	[657.28:10516.5]	[0.0038:0.0608]
	0.03	[599.37:5993.7]		[0.0085:0.1369]
water-Sand,	0.02	[2.4164:12.0824]	[0.2777:6.9444]	[5.7600:144.00]
water- GB ,		[2.5713:12.8867]		
water- AF		[2.5713:12.8867]		
water-Sand,	0.03	[1.0148:5.0741]	[340.556:3405.56]	[12.960:324.00]
water- GB ,		[1.0708:5.3543]		
water- AF		[1.0708:5.3543]		

**Table 3 entropy-26-00002-t003:** Asymptotic expressions of 
Ra
 in 
BBC
 and 
DBC
 cases with 
LTE
 assumption and with 
$m=\frac{a}{S_{o}}$
 (applicable to both 
CE
 and 
RE
).

Cases	Ra [2]
BBC	$\frac{{\left(m^{2}+\pi^{2}\right)}^{2}}{m^{2}}\left[\Lambda \left(m^{2}+\pi^{2}\right)+\sigma^{2}\right]$
DBC $\left(\sigma^{2}\gg 1\right)$	$\frac{{\left(m^{2}+\pi^{2}\right)}^{2}}{m^{2}}$

**Table 4 entropy-26-00002-t004:** First seven roots of the constraint 
$J_{\frac{\delta +1}{2}}\left[a_{n}\right]=0$
 in the cases of 
CE
 and 
RE
.

*n*	0	1	2	3	4	5	6
$a_{n}\left(\delta =1\right)$	3.83171	7.01559	10.1735	13.3237	16.4706	19.6159	22.7601
$a_{n}\left(\delta =0\right)$	0	π	2π	3π	4π	5π	6π

**Table 5 entropy-26-00002-t005:** Expression for the number of cells in the limiting cases of 
BBC
-
LTNE
.

	Parameters	Number of Cells
DBC (LTNE)	σ≫1	$n=\frac{1}{2\pi }\left[-\pi +\sqrt{\pi^{2}+2\sqrt{q_{11}+q_{12}}-2q_{3}S_{o}^{2}+q_{15}}\right]+q_{23}$
RBC	$\sigma^{2}=0$ and Λ=1	$n=\frac{-1}{2}+\sqrt{\frac{1}{4}-\frac{1}{6}\left[q_{10}\left(1-\frac{q_{10}}{q_{9}}\right)-q_{9}\right]}+q_{23}$
BBC (LTE)	H→0 or H→∞	$\frac{1}{2}\left[-1+\sqrt{1-\frac{2}{3\Lambda \pi^{2}}\left(q_{20}-\frac{2^{\frac{2}{3}}q_{20}^{2}+q_{22}^{2}}{2^{\frac{1}{3}}q_{22}}\right)}\right]+q_{23}$
DBC (LTE)	σ≫1 , H→0 or H→∞	$\frac{1}{2}\left[\sqrt{1+4S_{o}^{2}}-1\right]+q_{23}$

**Table 6 entropy-26-00002-t006:** Ranges of 
$S_{o}$
 at which the number of cells increases in different water-saturated porous media with 
H=5.0
 (in the case of 
LTNE
) and 
H=104
 (in the case of 
LTE
 except for 
AF
 for which we need to take 
H=107
), and 
d=0.03
 m (in the case of 
BBC
) (The values in bold are the threshold values referred to as 
$S_{o}^{T}$
 in Figure 2.

**Water-** pm	**Cases**	**Number of Cells (Cylindrical Enclosure)**							
		**1**	**2**	**3**	**4**	**5**	**6**	**7**	**8**
water- GF	BBC-LTNE	[0.68, 1.98)	[1.98, 3.19)	[3.19, 4.39)	[4.39, 5.58)	[5.58, 6.76)	[6.76, 7.95)	[7.95, 9.13)	[9.13, 10.32)
	BBC - LTE	[0.68, 1.99)	[1.99, 3.20)	[3.20, 4.40)	[4.40, 5.59)	[5.59, 6.78)	[6.78, 7.97)	[7.97, 9.15)	[9.15, 10.34)
water	RBC	[0.82, 2.37)	[2.37, 3.82)	[3.82, 5.25)	[5.25, 6.67)	[6.67, 8.09)	[8.09, 9.51)	[9.51, 10.92)	[10.92, 12.34)
water-sand	DBC-LTNE	[0.52, 1.59)	[1.59, 2.57)	[2.57, 3.54)	[3.54, 4.50)	[4.50, 5.46)	[5.46, 6.42)	[6.42, 7.38)	[7.38, 8.33)
	DBC-LTE	[0.54, 1.66)	[1.66, 2.69)	[2.69, 3.70)	[3.70, 4.71)	[4.71, 5.72)	[5.72, 6.72)	[6.72, 7.72)	[7.72, 8.72)
water- GB	DBC-LTNE	[0.5, 1.53)	[1.53, 2.48)	[2.48, 3.42)	[3.42, 4.34)	[4.34, 5.27)	[5.27, 6.20)	[6.20, 7.12)	[7.12, 8.04)
	DBC-LTE	[0.54, 1.66)	[1.66, 2.69)	[2.69, 3.70)	[3.70, 4.71)	[4.71, 5.72)	[5.72, 6.72)	[6.72, 7.72)	[7.72, 8.72)
water- AF	DBC-LTNE	[0.49, 1.50)	[1.50, 2.43)	[2.43, 3.34)	(3.34, 4.25)	[4.25, 5.16)	[5.16, 6.07)	[6.07, 6.97)	[6.97, 7.88)
	DBC-LTE	[0.54, 1.66)	[1.66, 2.69)	[2.69, 3.70)	[3.70, 4.71)	[4.71, 5.72)	[5.72, 6.72)	[6.72, 7.72)	[7.72, 8.72)
**Water**- pm	**Cases**	**Number of Cells (Rectangular Enclosure)**							
		**1**	**2**	**3**	**4**	**5**	**6**	**7**	**8**
water- GF	BBC-LTNE	(0, 1.70)	[1.70, 2.91)	[2.91, 4.11)	[4.11, 5.30)	[5.30, 6.49)	[6.49, 7.67)	[7.67, 8.86)	[8.86, 10.04)
	BBC - LTE	(0, 1.70)	[1.70, 2.92)	[2.92, 4.12)	[4.12, 5.31)	[5.31, 6.50)	[6.50, 7.69)	[7.69, 8.88)	[8.88, 10.06)
water	RBC	(0, 2.03)	[2.03, 3.48)	[3.48, 4.92)	[4.92, 6.34)	[6.34, 7.76)	[7.76, 9.18)	[9.18, 10.59)	[10.59, 12.01)
water-sand	DBC-LTNE	(0, 1.36)	[1.36, 2.34)	[2.34, 3.31)	[3.31, 4.28)	[4.28, 5.24)	[5.24, 6.20)	[6.20, 7.15)	[7.15, 8.11)
	DBC-LTE	(0, 1.42)	[1.42, 2.45)	[2.45, 3.47)	[3.47, 4.48)	[4.48, 5.48)	[5.48, 6.49)	[6.49, 7.49)	[7.49, 8.49)
water- GB	DBC-LTNE	(0, 1.31)	[1.31, 2.26)	[2.26, 3.20)	[3.20, 4.13)	[4.13, 5.06)	[5.06, 5.98)	[5.98, 6.91)	[6.91, 7.83)
	DBC-LTE	(0, 1.42)	[1.42, 2.45)	[2.45, 3.47)	[3.47, 4.48)	[4.48, 5.48)	[5.48, 6.49)	[6.49, 7.49)	[7.49, 8.49)
water- AF	DBC-LTNE	(0, 1.28)	[1.28, 2.22)	[2.22, 3.13)	[3.13, 4.04)	[4.04, 4.95)	[4.95, 5.86)	[5.86, 6.76)	[6.76, 7.66)
	DBC-LTE	(0, 1.42)	[1.42, 2.45)	[2.45, 3.47)	[3.47, 4.48)	[4.48, 5.48)	[5.48, 6.49)	[6.49, 7.49)	[7.49, 8.49)

**Table 7 entropy-26-00002-t007:** Cell-sizes of first 8 cells in different water-saturated porous media with 
H=5.0
 (in the case of 
LTNE
) and 
H=104
 (in the case of 
LTE
 except for 
AF
 for which we need to take 
H=107
), and 
d=0.03
 m (in the case of 
BBC
).

**Water-** pm	**Cases**	**Cell Aspect Ratios (Cylindrical Enclosure)**							
		**1**	**2**	**3**	**4**	**5**	**6**	**7**	**8**
water- GF	BBC-LTNE(d=0.003m)	1.30	1.21	1.20	1.19	1.18	1.19	1.18	1.19
	BBC-LTE(d=0.003m)	1.31	1.21	1.20	1.19	1.19	1.19	1.18	1.19
water	RBC	1.55	1.45	1.43	1.42	1.42	1.42	1.41	1.42
water-sand	DBC-LTNE	1.07	0.98	0.97	0.96	0.96	0.96	0.96	0.95
	DBC-LTE	1.12	1.03	1.01	1.01	1.01	1.00	1.00	1.00
water- GB	DBC-LTNE	1.03	0.95	0.94	0.92	0.93	0.93	0.92	0.92
	DBC-LTE	1.12	1.03	1.01	1.01	1.01	1.00	1.00	1.00
water- AF	DBC-LTNE	1.01	0.93	0.91	0.91	0.91	0.91	0.90	0.91
	DBC-LTE	1.12	1.03	1.01	1.01	1.01	1.00	1.00	1.00
**Water**- pm	**Cases**	**Cell Aspect Ratios (Rectangular Enclosure)**							
		**1**	**2**	**3**	**4**	**5**	**6**	**7**	**8**
water- GF	BBC-LTNE(d=0.03m)	1.21	1.20	1.19	1.19	1.18	1.19	1.18	1.19
	BBC-LTE(d=0.03m)	1.22	1.20	1.19	1.19	1.19	1.19	1.18	1.19
water	RBC	1.45	1.43	1.42	1.42	1.41	1.42	1.41	1.42
water-sand	DBC-LTNE	0.98	0.97	0.97	0.96	0.96	0.95	0.96	0.96
	DBC-LTE	1.03	1.02	1.01	1.00	1.01	1.00	1.00	1.00
water- GB	DBC-LTNE	0.95	0.94	0.93	0.93	0.92	0.93	0.92	0.92
	DBC-LTE	1.03	1.02	1.01	1.00	1.01	1.00	1.00	1.00
water- AF	DBC-LTNE	0.94	0.91	0.91	0.91	0.91	0.90	0.90	0.91
	DBC-LTE	1.03	1.02	1.01	1.00	1.01	1.00	1.00	1.00

**Table 8 entropy-26-00002-t008:** Expressions of 
$Ra_{c}$
 and *n* in different problems.

Cases	${\mathrm{Ra}}_{c}$	$n[S_{o}]$
BBC (LTNE)	$\frac{{\left[{\left(\delta \left(a_{2}-2\pi \right)+n\pi \right)}^{2}/S_{o}^{2}+\pi^{2}\right]}^{2}}{{\left(\delta \left(a_{2}-2\pi \right)+n\pi \right)}^{2}/S_{o}^{2}}\left[\Lambda \left[{\left(\delta \left(a_{2}-2\pi \right)+n\pi \right)}^{2}/S_{o}^{2}+\pi^{2}\right]+\sigma^{2}\right]\times$ $\left[1+\frac{H}{{\left(\delta \left(a_{2}-2\pi \right)+n\pi \right)}^{2}/S_{o}^{2}+\pi^{2}+H\gamma }\right]$	$\frac{-\pi +\sqrt{\pi^{2}+4Q}}{2\pi }+\delta \left[2-\frac{a_{2}}{\pi }\right]$
DBC (LTNE)	$\frac{{\left[{\left(\delta \left(a_{2}-2\pi \right)+n\pi \right)}^{2}/S_{o}^{2}+\pi^{2}\right]}^{2}}{{\left(\delta \left(a_{2}-2\pi \right)+n\pi \right)}^{2}/S_{o}^{2}}\times \left[1+\frac{H}{{\left(\delta \left(a_{2}-2\pi \right)+n\pi \right)}^{2}/S_{o}^{2}+\pi^{2}+H\gamma }\right]$	$\frac{-\pi +\sqrt{\pi^{2}+4Q}}{2\pi }+\delta \left[2-\frac{a_{2}}{\pi }\right]$
RBC	$\frac{{\left[{\left(\delta \left(a_{2}-2\pi \right)+n\pi \right)}^{2}/S_{o}^{2}+\pi^{2}\right]}^{3}}{{\left(\delta \left(a_{2}-2\pi \right)+n\pi \right)}^{2}/S_{o}^{2}}$	$n=\frac{-1}{2}+\sqrt{\frac{1}{4}-\frac{1}{6}\left[q_{10}\left(1-\frac{q_{10}}{q_{9}}\right)-q_{9}\right]}+\delta \left(2-\frac{a_{2}}{\pi }\right)$
BBC (LTE)	$\frac{{\left[{\left(\delta \left(a_{2}-2\pi \right)+n\pi \right)}^{2}/S_{o}^{2}+\pi^{2}\right]}^{2}}{{\left(\delta \left(a_{2}-2\pi \right)+n\pi \right)}^{2}/S_{o}^{2}}\left[\Lambda \left[{\left(\delta \left(a_{2}-2\pi \right)+n\pi \right)}^{2}/S_{o}^{2}+\pi^{2}\right]+\sigma^{2}\right]$	$\frac{1}{2}\left[-1+\sqrt{1-\frac{2}{3\Lambda \pi^{2}}\left(q_{20}-\frac{2^{\frac{2}{3}}q_{20}^{2}+q_{22}^{2}}{2^{\frac{1}{3}}q_{22}}\right)}\right]+\delta \left[2-\frac{a_{2}}{\pi }\right]$
DBC (LTE)	$\frac{{\left[{\left(\delta \left(a_{2}-2\pi \right)+n\pi \right)}^{2}/S_{o}^{2}+\pi^{2}\right]}^{2}}{{\left(\delta \left(a_{2}-2\pi \right)+n\pi \right)}^{2}/S_{o}^{2}}$	$\frac{1}{2}\left[\sqrt{1+4S_{o}^{2}}-1\right]+\delta \left[2-\frac{a_{2}}{\pi }\right]$

## Data Availability

No new data were created or analyzed in this study. Data sharing is not applicable to this article.

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
