# Peer review of "A Unified Approach to Two-Dimensional Brinkman-Bénard Convection of Newtonian Liquids in Cylindrical and Rectangular Enclosures"

_entropy, 2023, doi:10.3390/e26010002_

Round 1

Reviewer 1 Report

Comments and Suggestions for Authors

A) General comments:

1)  The paper is interesting and rigorously presented; its topic is up to date and of interest; The mathematical analysis is rigorous and complete, also the methodology for unifying the analysis is very interesting and provides an actual progress. Once the minor points I raise in the specific comments are addressed, I will have no doubt in recommending publication. 

B) Specific comments:

The comments are identified by paragraph/line number on left

1)      Par.1: Well done and complete, though there are some typos, please check carefully.

2)      Par. 2: Mathematical analysis complete and well described; the large number of parameters/symbols, although they are all defined in the text, suggests that a more organized list of symbols would be very useful.

3)      Line 387-388: It is very interesting that n=1 is not necessarily the value yielding critical Ra, but, how can one see this from Fig. 2? OK n are the modes of the equation, and with some afterthought one realizes that the plots in Fig. 2 are the superpositions of the plots of eq. 63 for the different modes by taking only the lowest value of Ra, but this must be explained more clearly. Also, although it is evident that the first part (up to the first SoT) corresponds to n=1 and so on, it wouldn’t hurt to say this explicitly, maybe by indicating the value of n corresponding to the different regions and maybe differentiating the branches of the plots using different colours. I insist on this point because I think that a clear comprehension of Fig. 2 is crucial to the understanding of the whole paper.

4)      Line 497: OK, but this table of abbreviations would be more useful at the beginning of the paper, not at the end……

Comments on the Quality of English Language

The English expression is in general OK, although there are quite a few typos and, here and there, a few articles appear to be missing; I suggest a careful proofread and check on the paper by the Authors to clear these minor points.

Author Response

RESPONSE TO REFEREE 1

Referee #1:

Manuscript Number: Entropy-2652304

Comments and Suggestions for Authors

  1. A) General comments:

  • The paper is interesting and rigorously presented; its topic is up to date and of interest; The mathematical analysis is rigorous and complete, also the methodology for unifying the analysis is very interesting and provides an actual progress. Once the minor points I raise in the specific comments are addressed, I will have no doubt in recommending publication. 

  1. B) Specific comments:

The comments are identified by paragraph/line number on left

  1. 1: Well done and complete, though there are some typos, please check carefully.

Author:

The typo errors are now corrected.

  1. 2: Mathematical analysis complete and well described; the large number of parameters/symbols, although they are all defined in the text, suggests that a more organized list of symbols would be very useful.

Author:

The nomenclature is now included in the paper(see page 25).

  1. Line 387-388: It is very interesting that n=1 is not necessarily the value yielding critical Ra, but, how can one see this from Fig. 2? OK n are the modes of the equation, and with some afterthought one realizes that the plots in Fig. 2 are the superpositions of the plots of eq. 63 for the different modes by taking only the lowest value of Ra, but this must be explained more clearly. Also, although it is evident that the first part (up to the first SoT) corresponds to n=1 and so on, it wouldn’t hurt to say this explicitly, maybe by indicating the value of n corresponding to the different regions and maybe differentiating the branches of the plots using different colours. I insist on this point because I think that a clear comprehension of Fig. 2 is crucial to the understanding of the whole paper.

Author:

The details are now included in the lines 387-390, page no. 14.

  1. Line 497: OK, but this table of abbreviations would be more useful at the beginning of the paper, not at the end……

Author:

All the parameters are now included in the nomenclature(see page 25).

Comments on the Quality of English Language

The English expression is in general OK, although there are quite a few typos and, here and there, a few articles appear to be missing; I suggest a careful proofread and check on the paper by the Authors to clear these minor points.

Author:

Typo errors are now corrected in the article.

The authors are grateful to the Referee for most useful comments.

Reviewer 2 Report

Comments and Suggestions for Authors

The manuscript entitled A unified approach to two-dimensional Brinkman-Benard convection of Newtonian liquids in cylindrical and rectangular enclosures is theoretical study of Brinkman-Benard convection. I commented as follows;

1.The validity of the scheme should be shown. The present results should be compared with the experimental results.

2.The author should separate results and discussions.

3.The many symbols and letters are used. The author should summarize them as a nomenclature.

Comments on the Quality of English Language

English is fine.

Author Response

RESPONSE TO REFEREE 2

Referee #2:

Manuscript Number: Entropy-2652304

Comments and Suggestions for Authors

The manuscript entitled A unified approach to two-dimensional Brinkman-Benard convection of Newtonian liquids in cylindrical and rectangular enclosures is theoretical study of Brinkman-Benard convection. I commented as follows;

1.The validity of the scheme should be shown. The present results should be compared with the experimental results.

Author:

The details are now included in the lines 494-495 and 498-504 in page no. 24.

2.The author should separate results and discussions.

Author:

In most of the existing articles of MDPI Entropy journal, the results and discussion part was unified. Hence the present article was written in a similar form. Kindly accept the article in this form.

3.The many symbols and letters are used. The author should summarize them as a nomenclature.

Author:

The nomenclature is now included in the paper(see page 25).

Comments on the Quality of English Language

English is fine.

The authors are grateful to the Referee for most useful comments.

Reviewer 3 Report

Comments and Suggestions for Authors

The paper presents an analysis of two-dimensional Brinkman-Benard / Rayleigh-Benard / Darcy-Benard convection in cylindrical and rectangular enclosures (CE/RE) saturated by a Newtonian liquid. On the basis of the literature review on Rayleigh-Benard convection in a fluid-saturated loosely packed porous medium and a densely-packed porous medium, the authors identified areas that require analytical studies. In my opinion, the analytical study carried out is new and constitutes an important contribution to the development of knowledge on the convection of Newtonian liquids in porous media. The work is written well. The goal was clearly defined, and the mathematical model and calculation results were discussed in detail. I believe the work can be published in its current form. However, the authors could have written the conclusion better. The conclusions provide a lot of detailed information. Personally, I think the conclusions should contain more general information about the achievements presented in the work.

Author Response

RESPONSE TO REFEREE 3

Referee #3:

Manuscript Number: Entropy-2652304

Comments and Suggestions for Authors

The paper presents an analysis of two-dimensional Brinkman-Benard / Rayleigh-Benard / Darcy-Benard convection in cylindrical and rectangular enclosures (CE/RE) saturated by a Newtonian liquid. On the basis of the literature review on Rayleigh-Benard convection in a fluid-saturated loosely packed porous medium and a densely-packed porous medium, the authors identified areas that require analytical studies. In my opinion, the analytical study carried out is new and constitutes an important contribution to the development of knowledge on the convection of Newtonian liquids in porous media. The work is written well. The goal was clearly defined, and the mathematical model and calculation results were discussed in detail. I believe the work can be published in its current form. However, the authors could have written the conclusion better. The conclusions provide a lot of detailed information. Personally, I think the conclusions should contain more general information about the achievements presented in the work.

Author:

Conclusion part is now updated, see page 24.

The authors are grateful to the Referee for most useful comments.
